# No Subclass Left Behind: Fine-Grained Robustness in Coarse-Grained Classification Problems

Nimit Sohoni[‡], Jared A. Dunnmon[†], Geoffrey Angus[†], Albert Gu[†], and Christopher Ré[†]

[†]Department of Computer Science, Stanford University
[‡]Institute for Computational and Mathematical Engineering, Stanford University
`{nims, jdunnmon, gdlangus, albertgu, chrismre}@cs.stanford.edu`

## Abstract

In real-world classification tasks, each class often comprises multiple finer-grained "subclasses." As the subclass labels are frequently unavailable, models trained using only the coarser-grained class labels often exhibit highly variable performance across different subclasses. This phenomenon, known as *hidden stratification*, has important consequences for models deployed in safety-critical applications such as medicine. We propose GEORGE, a method to both measure and mitigate hidden stratification *even when subclass labels are unknown*. We first observe that unlabeled subclasses are often separable in the feature space of deep models, and exploit this fact to estimate subclass labels for the training data via clustering techniques. We then use these approximate subclass labels as a form of noisy supervision in a distributionally robust optimization objective. We theoretically characterize the performance of GEORGE in terms of the worst-case generalization error across any subclass. We empirically validate GEORGE on a mix of real-world and benchmark image classification datasets, and show that our approach boosts worst-case subclass accuracy by up to 14 percentage points compared to standard training techniques, without requiring any information about the subclasses.

## 1 Introduction

In many real-world classification tasks, each labeled class consists of multiple semantically distinct subclasses that are unlabeled. Because models are typically trained to maximize *global* metrics such as average performance, they often underperform on important subclasses [52, 40]. This phenomenon—recently termed *hidden stratification*—can lead to skewed assessments of model quality and result in unexpectedly poor performance when models are deployed [36]. For instance, a medical imaging model trained to classify between benign and abnormal lesions may achieve high overall performance, yet consistently mislabel a rare but critical abnormal subclass as "benign" [17].

Modern robust optimization techniques can improve performance on poorly-performing groups when the group identities are known [43]. However, in practice, a key obstacle is that *subclasses are often unlabeled*, or even unidentified. This makes even detecting such performance gaps—let alone mitigating them—a challenging problem. Nevertheless, recent empirical evidence [36] encouragingly suggests that feature representations of deep neural networks often carry information about unlabeled subclasses (see Figure 1). Motivated by this observation, we propose a method for addressing hidden stratification, by both measuring and improving worst-case subclass performance in the setting where subclass labels are unavailable. Our work towards this is organized into four main sections.

First, in Section 3 we propose a simple generative model of the data labeling process. Using this model, we show that when label annotations are insufficiently fine-grained—as is often the case in real-world datasets—hidden stratification can naturally arise. For instance, an image classification task might be to classify birds vs. frogs; if labels are only provided for these broad classes, they may

fail to capture visually meaningful finer-grained, intra-class variation (e.g., "bird in flight" versus "bird in nest"). We show that in the setting of our generative model, standard training via empirical risk minimization (ERM) can result in arbitrarily poor performance on underrepresented subclasses.

Second, in Section 4 we use insights from this generative model to motivate GEORGE, a two-step procedure for alleviating hidden stratification by first *estimating* the subclass labels and then *exploiting* these estimates to train a robust classifier. To estimate subclass labels, we train a standard model on the task, and split each class (or "superclass," for clarity) into estimated subclasses via unsupervised clustering in the model's feature space. We then exploit these estimated subclasses by training a new model to optimize *worst-case* performance over all estimated subclasses using group distributionally robust optimization (GDRO [43]). In this way, our framework allows ML practitioners to automatically detect poorly-performing subclasses and improve performance on them, without needing to resort to expensive manual relabeling of the data.

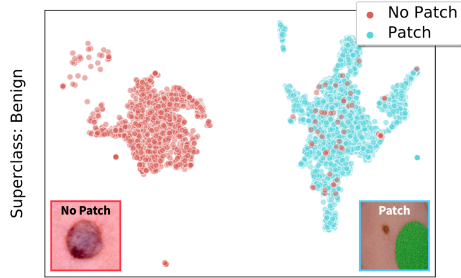

Figure 1: Benign class examples in the feature space of a model classfying skin lesions as benign or malignant. Benign examples containing a brightly colored patch (blue) and those without a patch (red) are separable in model feature space, even though the labels do not specify the presence of patches.

Third, in Section 5 we use our generative framework to prove that—under conditions on the data distribution and the quality of the recovered clusters— GEORGE can reduce the subclass performance gap, attaining the same asymptotic sample complexity rates as if the true subclass labels were known.

Fourth, in Section 6 we empirically validate the ability of GEORGE to both *measure* and *mitigate* hidden stratification on four image classification tasks, comprising both robustness benchmarks and real-world datasets. We show that the first step of GEORGE—training an ERM model and clustering the superclass features—often recovers clusters that align closely with true subclasses. We evaluate the ability of these clusters to measure the worst-case subclass (i.e., "robust") performance: on average, the gap between worst-case cluster performance and worst-case subclass performance is less than 40% of the gap between overall and worst-case subclass performance, indicating that GEORGE enables more accurate measurement of robust performance. Next, we show that the second stage of GEORGE—retraining a robust model using cluster assignments as proxy subclass labels— reduces average worst-case subclass error rates by over 23%. For comparison, the state-of-the-art "oracle" GDRO method that *does* require subclass labels [43] reduces average worst-case subclass error rates by 50%. As an extension, we show that leveraging recent pretrained image embeddings [27] for clustering can substantially further improve the robust performance of GEORGE, in some cases to match the performance of GDRO trained using the true subclass labels.

## 2   Background

### 2.1   Related Work

Our work builds upon prior work from three main areas: robust optimization, representation learning, and unsupervised clustering. We provide a more extensive discussion of related work in Appendix A.

**Distributionally Robust Optimization** Robustness and fairness is an active research area in machine learning [4, 21, 30, 26]. *Distributionally robust optimization* (DRO) attempts to guarantee good performance in the presence of distribution shift, e.g., from adversarial perturbations [49, 47] or evaluation on arbitrary subpopulations [16]. Because these notions of robustness can be pessimistic [23], others investigate *group* DRO (GDRO), which optimizes worst-case performance over a known set of subgroups [23, 43]. A major obstacle to applying GDRO methods in practice is that subgroup labels are often unavailable; in our work, we aim to address this issue in the classification setting.

**Representation Learning & Clustering** Our approach relies on estimating unknown subclass labels by clustering a feature representation of the data. Techniques for learning semantically useful image features include autoencoder-based methods [32, 46], the use of unsupervised auxiliary tasks [2, 9], and pretraining on massive datasets [27]. Such features may be used for unsupervised identification

of classes, either using clustering techniques [6] or in an end-to-end approach [25, 18]. It has also been observed that when a model is trained on coarse-grained class labels, the data *within each class* can often be separated into distinct clusters in model feature space [36]. While we focus on the latter approach, we also evaluate the utility of pretrained embeddings as a source of features for clustering.

## 2.2 Problem Setup

We are given $n$ datapoints $x_1, \ldots, x_n \in \mathcal{X}$ and associated *superclass* labels $y_1, \ldots, y_n \in \{1, \ldots, B\}$. In addition, associated with each datapoint $x_i$ is a latent (unobserved) *subclass* label $z_i \in \{1, \ldots, C\}$. We assume that $\{1, \ldots, C\}$ is partitioned into disjoint sets $S_1, \ldots, S_B$ such that if $z_i \in S_b$, then $y_i = b$; in other words, the subclass label $z_i$ determines the superclass label $y_i$. Let $S_b$ denote the set of all subclasses comprising superclass $b$, and $S(c)$ denote the superclass corresponding to subclass $c$.

Our goal is to classify examples from $\mathcal{X}$ into their correct *superclass*. Given a function class $\mathcal{F}$, it is typical to seek a classifier $f \in \mathcal{F}$ that maximizes overall population accuracy:
$$\underset{f \in \mathcal{F}}{\operatorname{argmax}} \, \mathbb{E}_{(x,y)} \left[ \mathbf{1}(f(x) = y) \right] . \tag{1}$$

By contrast, we seek to maximize the *robust accuracy*, defined as the *worst-case* expected accuracy over all subclasses:
$$\underset{f \in \mathcal{F}}{\operatorname{argmax}} \, \underset{c \in \{1, \ldots, C\}}{\min} \, \mathbb{E}_{(x,y)|z=c} \left[ \mathbf{1}(f(x) = y) \right] . \tag{2}$$

Note that $y$ is fixed conditional on the value of $z$. As we cannot directly optimize the population accuracy, we select a surrogate loss function $\ell$ and attempt to minimize this loss over the training data. For instance, the standard ERM approach to approximate (1) minimizes the empirical risk $R(f)$:

$$\underset{f \in \mathcal{F}}{\operatorname{argmin}} \left\{ R(f) := \frac{1}{n} \sum_{i=1}^{n} \ell(f(x_i), y_i) \right\} . \tag{3}$$

To approximate (2), if we knew $z_1, \ldots, z_n$ we could minimize the *worst-case per-subclass training risk* by solving:
$$\underset{f \in \mathcal{F}}{\operatorname{argmin}} \left\{ R_{\text{robust}}(f) := \underset{c \in \{1, \ldots, C\}}{\max} \frac{1}{n_c} \sum_{i=1}^{n} \mathbf{1}(z_i = c) \ell(f(x_i), y_i) \right\}, \tag{4}$$

where $n_c = \sum_{i=1}^{n} \mathbf{1}(z_i = c)$ is the number of training examples from subclass $c$. $R_{\text{robust}}(f)$ is the "robust loss" achieved by $f$. Our goal is to learn a model $\tilde{f} \in \mathcal{F}$ such that $R_{\text{robust}}(\tilde{f}) - \underset{f \in \mathcal{F}}{\min}(R_{\text{robust}}(f))$ is small with high probability. When the $z_i$'s are known, Eq. (4) can be tractably optimized using group distributionally robust optimization (GDRO) [23, 43]. However, we do *not* assume access to the $z_i$'s; we seek to approximately minimize $R_{\text{robust}}$ without knowledge of the subclass labels.

## 3 Modeling Hidden Stratification

In Section 3.1, we introduce a generative model of the data labeling process. In Section 3.2, we use this model to explain how hidden stratification can occur, and show that in the setting of this model ERM can attain arbitrarily poor robust risk compared to DRO.

### 3.1 A Model of the Data Generating and Labeling Process

In real datasets, individual datapoints are typically described by multiple different attributes, yet often only a subset of these are captured by the class labels. For example, a dataset might consist of images labeled "cat" or "dog." These coarse class labels may not capture other salient attributes (color, size, breed, etc.); these attributes can be interpreted as latent variables representing different subclasses.

We model this phenomenon with a hierarchical data generation process. First, a binary vector $\vec{Z} \in \{-1, +1\}^k$ is sampled from a distribution $p(\vec{Z})$. Each entry $Z_i$ is an attribute, while each unique value of $\vec{Z}$ represents a different subclass. Then, a latent "feature vector" $\vec{V} \in \mathbb{R}^k$ is sampled from a distribution conditioned on $\vec{Z}$, where, when conditioned on $Z_i$, each individual feature $V_i$ is Gaussian and independent of $Z_j$ for $j > i$. Finally, the datapoint $X \in \mathcal{X}$ is determined by the latent features $\vec{V}$ via a fixed map $g : \mathbb{R}^k \to \mathcal{X}$. Meanwhile, the superclass label $Y$ is a fixed discrete-valued

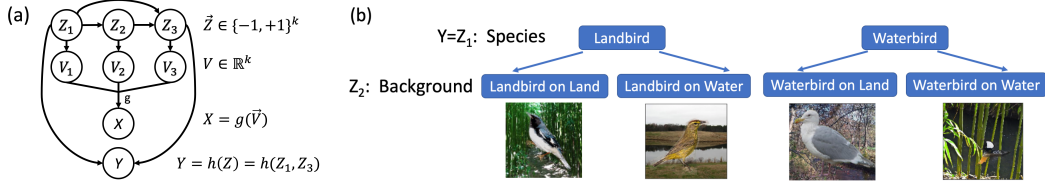

Figure 2: (a) Generative model of hidden stratification: attributes $Z$ determine features $\vec{V}$ and labels $Y$; mapping $g$ transforms $\vec{V}$ to yield observed data $X$. (b) On the Waterbirds dataset [43], attributes $(Z_1, Z_2)$ denote species and background type respectively; the label $Y$ is the species type.

function $h(\vec{Z})$ of $\vec{Z}$. In particular, $h$ may only depend on a subset of the $Z_i$'s; the $Z_i$'s which do not influence the label $Y$ correspond to hidden subclasses. $X, Y$ are observed, while $\vec{V}, \vec{Z}$ are not. Fig. 2a illustrates this generative process; Fig. 2b presents an analogue on the Waterbirds dataset [43].

Importantly, rather than attempting to enforce good performance on *all possible subsets* of the data, we assume some meaningful structure to the subclasses. We model this via the Gaussian assumption on $p(V_i|\vec{Z})$, which is similar to that often made for the latent space of GANs [5]. Consequently, the data distribution is a mixture of Gaussians in "feature space," which facilitates further theoretical analysis (Section 5). Our generative model also bears similarity to that of [23], who use a hierarchical data-generation model to analyze the behavior of DRO methods in the presence of distribution shift.

## 3.2 What Causes Hidden Stratification, and When Can It Be Fixed?

We now use our generative model to help understand why hidden stratification can occur, and present a simple example in which ERM is provably suboptimal in terms of the robust risk.

We distinguish between two main causes of hidden stratification: *inherent hardness* and *dataset imbalance*. First, certain subclasses are "inherently harder" to classify because they are more similar to other superclasses. We define the inherent hardness of a task as the minimum attainable robust error; inherent hardness thus lower bounds the worst-case subclass error of any model. See Appendix D for more discussion.

Second, imbalance in subclass sizes can cause ERM to underserve rare subclasses, since it optimizes for average-case performance. We provide a simple concrete example (3.1) below. Unlike inherent hardness, robust performance gaps arising from dataset imbalances *can be resolved* if subclass labels are known, by using these labels to minimize Eq. (4) via GDRO.

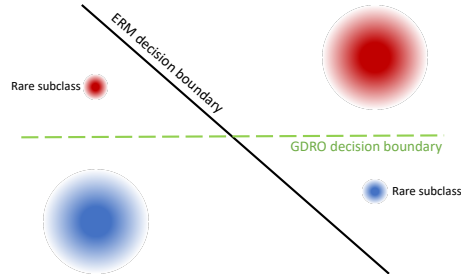

Figure 3: As $\alpha \to 0$ in Example 3.1, top-left & lower-right subclasses get rarer and are misclassified by ERM (black boundary), whereas GDRO learns the optimal robust boundary (green) to classify red vs. blue superclasses.

**Example 3.1** Fig. 3 depicts an example distribution generated by the model in Sec. 3.1. In this example, the binary attribute vector $\vec{Z}$ has dimension 2, i.e., $\vec{Z} = (Z_1, Z_2)$, while only $Z_2$ determines the superclass label $Y$, i.e., $Y = Z_2$. The latent attribute $Z_1$ induces two subclasses in each superclass, each distributed as a different Gaussian in feature space, with mixture proportions $\alpha$ and $1 - \alpha$ respectively. (See Appendix D.1 for the specific parameters of the per-subclass distributions in this example.) For linear models with regularized logistic loss, as the proportion $\alpha$ of the rare subclasses goes to 0, the worst-case subclass accuracy of ERM is only $O(\alpha)$, while that of GDRO is $1 - O(\alpha)$. (Proof in Appendix D.1.)

Example 3.1 illustrates that when the dataset is imbalanced—i.e., the distribution of the underlying attributes $\vec{Z}$ is highly nonuniform—knowledge of subclass labels can improve robust performance. We thus ask: *how well can we estimate subclass labels if they are not provided*? In the extreme, if two subclasses of a superclass have the same distribution in feature space, we cannot distinguish them. However, the model must then perform the same on each subclass, since its prediction is a fixed function of the features! Conversely, if one subclass has higher average error, it must lie "further across" the decision boundary, meaning that the two subclasses must be separable to a degree; the larger the accuracy gap, the more separable the subclasses are. We formalize this in Appendix D.3.

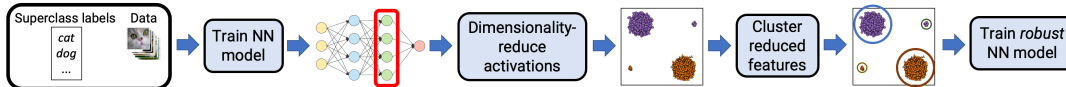

Figure 4: Schematic describing GEORGE. The inputs are the datapoints and superclass labels. First, a model is trained with ERM on the superclass classification task. The activations of the penultimate layer are then dimensionality-reduced, and clustering is applied to the resulting features to obtain estimated subclasses. Finally, a new model is trained using these clusters as groups for GDRO.

## 4 GEORGE: A Framework for Mitigating Hidden Stratification

Inspired by the insights of Sec. 3, we propose GEORGE, an algorithm to mitigate hidden stratification. A schematic overview of GEORGE is provided in Figure 4.

Under the generative model of Section 3.1, each subclass is described by a different Gaussian in latent feature space. This suggests that a natural approach to *identify* the subclasses is to transform the data into feature space, and then cluster the data into estimated subclasses. To obtain this feature space, we leverage the empirical observation that feature representations of deep neural networks trained on a *superclass* task can carry information about unlabeled *subclasses* [36]. Next, to *improve performance* on these estimated subclasses, we minimize the maximum *per-cluster* average loss, by using the clusters as groups in the GDRO objective [43]. We provide more details below, and detailed pseudocode in Appendix B (Algorithm 1).

### 4.1 Step 1: Estimating Approximate Subclass Labels

In the first step of GEORGE, we train an ERM model on the superclass task and cluster the feature representations of each superclass to generate proxy subclass labels. Formally, we train a deep neural network $L \circ f_\theta$ to predict the superclass labels, where $f_\theta : \mathcal{X} \to \mathbb{R}^d$ is a parametrized "featurizer" and $L : \mathbb{R}^d \to \Delta^B$ outputs classification logits. We then cluster the features output by $f_\theta$ for the data of each superclass into $k$ clusters, where $k$ is chosen automatically. To each datapoint $x_i$ in the training and validation sets, we associate its cluster assignment $\tilde{z}_i \in \{1, \ldots, k\}$; we use the $\tilde{z}_i$'s as surrogates for the true subclass labels $z_i$.

#### 4.1.1 Clustering Details

In practice, we apply UMAP dimensionality reduction [33] before clustering, as we find it improves results (Appendix B). Additionally, based on the insight of Section 3.2 that subclasses with high loss differences are more separable, we also use the loss component (i.e., the component of the activation vector orthogonal to the decision boundary) as an alternative representation.

We first tried using standard clustering methods (such as $k$-means and Gaussian mixture model clustering) in our work. By visual inspection, we found that these methods often failed to capture smaller clusters, even if they were well-separated. However, missing small clusters like this is problematic for GEORGE, since these small clusters frequently correspond to rare, low-performing subclass. Additionally, these methods require specification of $k$. We apply *over-clustering* (clustering using a larger $k$) to remedy this problem in an efficient manner. Naive overclustering also has drawbacks as it still requires manual specification of $k$, and if $k$ is set too large, several clusters can be spurious and result in overly pessimistic and unstable measurements of robust performance (as we explore in Appendix C.2.6). Thus, we develop a fully automated criterion based on the commonly used Silhouette (SIL) criterion [42] to search for the number of clusters $k$, over-cluster to find smaller clusters that were missed, and filter out the spurious overclusters. Empirically, our clustering approach significantly improves performance over "vanilla" clustering; we hope that it may be of independent interest as well. We describe our procedures in more detail in Appendix B.

$k$ and other clustering and representation hyperparameters are selected automatically based on an unsupervised SIL criterion [42] as described further in Appendix B.

### 4.2 Step 2: Exploiting Approximate Subclass Labels

In the second step of GEORGE, we use the GDRO algorithm from [43] and our estimated subclass labels $\tilde{z}_i$ to train a new classifier with better worst-case performance on the estimated subclasses.

Given data $\{(x_i, y_i, t_i)\}_{i=1}^n$ and loss function $\ell$, GDRO minimizes $\max\limits_{t \in \mathcal{T}} \mathbb{E}\limits_{x,y \sim \hat{P}_t} [\ell((L \circ f_\theta)(x), y)]$

w.r.t. parameters $(L, \theta)$, where $\mathcal{T}$ is the discrete set of groups and $\hat{P}_t$ is the empirical distribution of examples from group $t$. This coincides with the true objective (4) when the true subclass labels $z_i$ are used as the group labels $t_i$. In our case, we use the cluster assignments $\tilde{z}_i$ as the group labels instead, i.e., minimize $\max\limits_{1 \leq \tilde{z} \leq k} \mathbb{E}\limits_{x,y \sim \hat{P}_{\tilde{z}}} [\ell(L \circ f_\theta)(x), y)]$. In Appendix D, we present an extension to the GDRO algorithm of [43] to handle the case where the group assignments $\tilde{z}_i$ can be probabilistic labels in $\Delta^k$, instead of hard labels in $\{1, \ldots, k\}$.

## 5 Analysis of GEORGE

We now analyze a simple mixture model data distribution, based on the generative model in Section 3.1. We show that in this setting, unlike ERM, GEORGE converges to the optimal robust risk at the same sample complexity rate as GDRO when it is able to recover the true latent features $\vec{Z}$. Specifically, Example 3.1 shows that the robust risk of ERM can be arbitrarily worse than that of GDRO for data generated according to the generative model in Section 3.1. By contrast, if the subclass labels estimated by GEORGE are sufficiently accurate, then the objective minimized in Step 2 of GEORGE well approximates the true GDRO objective (4). In Theorem 1, we use this to show that, when each subclass is described by a different Gaussian in feature space, GEORGE achieves the same optimal asymptotic sample complexity rates as GDRO trained with true subclass labels. We sketch the argument below; full proofs are deferred to Appendix D.

First, suppose we could compute the true data distribution $\mathcal{P}(x, y, z)$. Our goal is to minimize the maximum per-subclass loss by solving Eq. (4). Even with infinite data, we cannot estimate the *individual* $z_i$'s to arbitrary accuracy, so we cannot directly compute the objective in (4). However, we can *estimate* the per-subclass losses as follows: for each example $(x_i, y_i)$, we use $\mathcal{P}$ to compute the probability that it came from subclass $c$, and use that to weight the loss corresponding to that example. In Lemma 1, we show that this yields an unbiased estimate of the average per-subclass empirical risk.

**Lemma 1.** *Let $R_c$ be the sample average loss of examples in subclass $c$. Let $w(x, c) := \frac{\mathcal{P}(x|z=c)}{\mathcal{P}(x|y=S(c))}$. Let $\tilde{R}_c$ be the sample average of $w(x_i, c)\ell(f(x_i), y_i)$ over all examples $x_i$ with superclass label $y_i = S(c)$. Then $\tilde{R}_c$ is an unbiased estimate of $R_c$, and their difference converges to 0 at $O(1/\sqrt{n})$.*

In practice, we do not have access to $\mathcal{P}$ and estimate it with $\hat{\mathcal{P}}$, computed from data. Thus, the weights $w(x, c)$ are replaced by weights $\hat{w}(x, c)$ estimated from $\hat{\mathcal{P}}$, leading to an estimate $\hat{R}_c$ of $\tilde{R}_c$. Nevertheless, if we can bound the total variation estimation error of $\hat{\mathcal{P}}$, we can use this to bound the error in this loss estimate, as shown in Lemma 2 (Appendix D). In Theorem 1, we leverage Lemma 2 and recent results on learning Gaussian mixtures [3] to show that, when each subclass is described by a different Gaussian, $\hat{\mathcal{P}}$ can be estimated well enough so that the minimizer of the perturbed robust loss converges to the minimizer of the true robust loss at the optimal sample complexity rate.

**Theorem 1.** *Let $\hat{R}_{robust} := \max_c \hat{R}_c$. Suppose $\ell, f$ are Lipschitz, $f$ has bounded parameters, and $\mathcal{P}(x|z=c)$ is Gaussian and unique for each subclass $c$. Then, if we estimate $\hat{\mathcal{P}}$ using the algorithm from [3], $\hat{f} := \min\limits_{f \in \mathcal{F}} \hat{R}_{robust}(f)$ satisfies $R_{robust}(\hat{f}) - \min\limits_{f \in \mathcal{F}} R_{robust}(f) \leq \tilde{O}(\sqrt{1/n})$ w.h.p.*

Theorem 1 implies that if each subclass is Gaussian in feature space, and we have access to this feature space (i.e., we can invert the mapping $g$ from features $\vec{Z}$ to data $X$), we can cluster the features to estimate $\hat{\mathcal{P}}$, and the robust generalization performance of the model that minimizes the resulting perturbed training loss $\hat{R}_{robust}$ scales the same as does that of the minimizer of the true robust training loss $R_{robust}$, in terms of the amount of data required. This underscores the importance of recovering a "good" feature space; empirically, we show in Appendix C that the choice of model architecture can indeed dramatically impact the model feature space and thus the ability to recover subclasses.

## 6 Experiments

We empirically validate that GEORGE can mitigate hidden stratification across four datasets. In Section 6.2, we show that when subclass labels are unavailable, GEORGE improves robust performance

Table 1: Robust and overall performance for ERM, GEORGE, and subclass-GDRO (i.e., GDRO with true subclass labels). Performance metric is accuracy for all datasets but ISIC, which uses AUROC. Bolded values are best between ERM and GEORGE, which do not require subclass labels. Sub-columns for ISIC represent two different definitions of the ISIC subclasses; see Section 6.3.

| Method | Requires Subclass Labels? | Metric Type | Waterbirds | U-MNIST | ISIC | | CelebA |
|---|---|---|---|---|---|---|---|
| | | | | | Non-patch | Histopath. | |
| ERM | ✗ | Robust | 63.3($\pm$1.6) | 93.9($\pm$0.6) | **.920**($\pm$.007) | .872($\pm$.010) | 41.1($\pm$2.3) |
| | | Overall | 97.2($\pm$0.1) | 98.2($\pm$0.1) | .956($\pm$.003) | | 95.7($\pm$0.1) |
| GEORGE (ours) | ✗ | Robust | **76.2**($\pm$2.0) | **95.7**($\pm$0.6) | .918($\pm$.009) | **.881**($\pm$.005) | **52.4**($\pm$1.3) |
| | | Overall | 95.5($\pm$0.6) | 97.9($\pm$0.2) | .935($\pm$.007) | | 94.8($\pm$0.2) |
| Subclass-GDRO | ✓ | Robust | 90.7($\pm$0.4) | 96.8($\pm$0.4) | .922($\pm$.007) | .876($\pm$.005) | 85.9($\pm$2.5) |
| | | Overall | 92.0($\pm$0.4) | 98.0($\pm$0.3) | .934($\pm$.010) | | 93.6($\pm$0.2) |

over standard methods. In Section 6.3, we analyze the clusters returned by GEORGE to understand the reasons for this improvement; we confirm that GEORGE identifies clusters that correspond to poorly-performing subclasses, which enables accurate measurement of robust performance. In Section 6.4, we evaluate the use of recent pretrained image embeddings [27] as a source of features for GEORGE, and find that this further improves performance of GEORGE on some applications. Additional details on datasets, model architectures, and experimental procedures are in Appendix B.

## 6.1  Datasets

**Waterbirds** Waterbirds, a robustness benchmark introduced to evaluate GDRO in [43], contains images of land-bird and water-bird species on either land or water backgrounds. The task is to classify images into land-bird vs. water-bird; however, 95% of land (water)-birds are on land (water) backgrounds, causing ERM to often misclassify both land-birds on water and water-birds on land.

**Undersampled MNIST (U-MNIST)** We design U-MNIST as a modified version of MNIST [28], where the task is to classify digits as "<5" and "≥5" (digits 0-9 are the subclasses). In addition, we remove 95% of '8's; due to its rarity, it is challenging for ERM to perform well on the '8' subclass.

**CelebA** CelebA is a common face classification dataset also used as a robustness benchmark in [43]. The task is to classify faces as "blond" or "not blond." Because only 6% of blond faces are male, ERM performs poorly on this rare subclass.

**ISIC** The ISIC skin cancer dataset [12] is a public real-world dataset for classifying skin lesions as "malignant" or "benign." 48% of benign images contain a colored patch. Of the non-patch examples, 49% required histopathology (a biopsy) to diagnose. We report AUROC for ISIC, as is standard [41].

## 6.2  End-to-End Results

We first show that GEORGE substantially improves the worst-case subclass accuracy, while modestly affecting overall accuracy. (Recall that we refer to worst-case subclass accuracy as "robust accuracy" [Eq. (2)].) We train models on each dataset in Sec. 6.1 using (a) ERM, (b) GEORGE, and (c) GDRO with true subclass labels ("subclass-GDRO"), and report both robust and overall performance metrics in Table 1. Compared to ERM, training with GEORGE improves robust accuracy by up to 14 points, and closes up to 62% of the gap between the robust error of the ERM model and that of the subclass-GDRO model—despite the fact that GEORGE does not require subclass labels.[1]

On Waterbirds, U-MNIST, and CelebA, GEORGE significantly improves worst-case subclass accuracy over ERM. On ISIC, GEORGE increases the clinically meaningful histopathology subclass AUROC over ERM, while all methods perform similarly on the non-patch subclass. On CelebA, although GEORGE improves upon ERM, it substantially underperforms subclass-GDRO. However, this gap can be closed when improved features are used: if we cluster pretrained BiT embeddings [27] rather than ERM features and use the resulting cluster assignments for the second stage of GEORGE, the robust accuracy improves to match that of subclass-GDRO. We describe this experiment in Sec. 6.4.

In terms of overall performance, ERM generally performs best (as it is designed to optimize for average-case performance), followed by GEORGE and then subclass-GDRO. However, this difference is generally much smaller in magnitude (less than 5 points) than the increase in robust performance.

## 6.3   Clustering Results

Step 1 of GEORGE is to train an ERM model and cluster the data of each superclass in its feature space. We analyze these clusters to better understand GEORGE's behavior. First, we show that GEORGE finds clusters that align well with poorly-performing human-labeled subclasses. This helps explain why the second step of GEORGE, running GDRO using the cluster assignments as groups, improves performance on these subclasses (as in Section 6.2). Next, we show that GEORGE can discover meaningful subclasses that were not labeled by human annotators. Finally, we show that the worst-case performance measured on the clusters returned by GEORGE is a good approximation of the true robust performance.

**Subclass Recovery** We evaluate the ability of GEORGE to identify clusters that correspond to the true subclasses. We focus on identification of *poorly-performing* subclasses, as these determine robust performance. In Table 2, we compute the precision and recall of the cluster returned by GEORGE that most closely aligns with each given subclass. *Precision* is the fraction of cluster examples with that subclass label; *recall* is the fraction of subclass examples assigned to the cluster. For each poorly-performing subclass, GEORGE identifies a cluster with high recall and better-than-random precision.

We note that the lower recall on ISIC is because the no-patch subclass is often split into two clusters; in fact, this subclass is actually composed of two semantically distinct groups as discussed below. If these two clusters are combined, their precision and recall at identifying no-patch examples is $\geq 0.99$.

**Unlabeled Subclass Discovery** In addition to yielding clusters aligned with human-annotated subclasses, our procedure can identify semantically meaningful subclasses that were not specified in the human-provided schema. On U-MNIST, 60% of trials of GEORGE partition the "7" subclass into two subclusters, each containing stylistically different images (Fig. 8c, App. C). On ISIC, 60% of GEORGE trials reveal two distinct benign clusters within the no-patch subclass (see Fig. 8h). In these trials, 77% of images in one no-patch cluster required histopathology (biopsy & pathologist referral), while such images made up <7% of the other cluster. In other words, the no-patch subclass split into "histopathology" and "non-histopathology" clusters, where the former datapoints were harder for clinicians to classify. We comment on the real-world importance of this result in Broader Impacts.

Table 2: Alignment of clusters with poorly-performing subclasses on the train set. We run Step 1 of GEORGE over multiple random seeds (i.e., train multiple ERM models and cluster their activations). In col. 4, we report the percentage of these trials with a cluster above the given precision and recall thresholds (cols. 5, 6) for identifying the subclass in col. 2. We report the proportion of examples from that subclass within its superclass in col. 3.

| Task | Subclass | Subclass Prevalence | % of trials | Precision | Recall |
|------|----------|---------------------|-------------|-----------|--------|
| U-MNIST | "8" digit | 0.012 | 100 | 0.81 | 0.80 |
| Waterbirds | Water-birds on land | 0.05 | 100 | 0.15 | 0.95 |
| Waterbirds | Land-birds on water | 0.05 | 100 | 0.30 | 0.93 |
| ISIC | No-patch | 0.48 | 100 | 0.99 | 0.60 |
| CelebA | blond males | 0.06 | 100 | 0.13 | 0.90 |

**Estimating Robust Accuracy** We show that the clusters returned by GEORGE enable improved measurement of worst-case subclass performance. Specifically, we measure the worst-case performance across any cluster returned by GEORGE (which we call the "cluster-robust" performance) and compare this to the true robust performance and the overall performance. We present results for both ERM and GEORGE in Table 3. In most cases, the cluster-robust performance is much closer to the true robust performance than the overall performance is. On ISIC, cluster-robust performance even yields a better estimate of robust performance on the histopathology subclass than does performance on the patch/no-patch subclass labels. By comparing cluster-robust performance to overall performance, we can detect hidden stratification (and estimate its magnitude) without requiring subclass labels.

Table 3: Comparison of overall, cluster-robust, and robust performance.[2](Conventions as in Table 1.)

| Method | Metric Type | Waterbirds | U-MNIST | ISIC Non-patch | Histopath. | CelebA |
|---|---|---|---|---|---|---|
| ERM | Robust | 63.3(±1.6) | 93.9(±0.6) | .920(±.007) | .872(±.010) | 41.1(±2.3) |
| | Cluster-Robust | 76.8(±1.6) | 92.3(±2.5) | .894(±.031) | | 59.1(±1.1) |
| | Overall | 97.3(±0.1) | 98.2(±0.1) | .956(±.003) | | 95.7(±0.1) |
| GEORGE | Robust | 76.2(±2.0) | 95.7(±0.6) | .918(±.009) | .881(±.005) | 52.4(±1.3) |
| | Cluster-Robust | 93.5(±0.5) | 93.5(±1.9) | .904(±.020) | | 71.8(±0.2) |
| | Overall | 95.5(±0.6) | 97.9(±0.2) | .935(±.007) | | 94.8(±0.2) |

In addition, improvements in robust performance from GEORGE compared to ERM are accompanied by increases in cluster-robust performance; by comparing the cluster-robust performance of ERM and GEORGE, we can estimate how much GEORGE improves hidden stratification.

## 6.4 Extension: Leveraging Pretrained Embeddings

As an alternative to training an ERM model, we assess whether recent pretrained image embeddings (BiT, [27]) can provide better features for Step 1 of GEORGE. Specifically, we modify Step 1 of GEORGE to compute BiT embeddings for the datapoints, cluster these, and use the cluster assignments as estimated subclass labels in Step 2 of GEORGE. This modification dramatically improves robust accuracy on CelebA to **86.0**%, matching subclass-GDRO. The CelebA BiT clusters align much better with the true subclasses, which helps explain this (e.g., for the "blond male" subclass, precision improves to 0.15 and recall to 0.98). However, GEORGE with BiT clustering performs worse than the default GEORGE implementation on Waterbirds, suggesting that BiT is not a panacea; on Waterbirds, the task-specific information contained in the representation of the trained ERM model is important for identifying meaningful clusters. See App. B.3.4 for additional evaluations and discussion. Extending Step 1 of GEORGE to enable automatically selecting between distinct representations (e.g., BiT vs. ERM) is a compelling future topic.

## 7 Conclusion

We propose GEORGE, a two-step approach for measuring and mitigating hidden stratification without requiring access to subclass labels. GEORGE's first step, clustering the features of an ERM model, identifies clusters that provide useful approximations of worst-case subclass performance. GEORGE's second step, using these cluster assignments as groups in GDRO, yields significant improvements in worst-case subclass performance. We analyze GEORGE in the context of a simple generative model, and show that under suitable assumptions GEORGE achieves the same asymptotic sample complexity rates as if we had access to true subclass labels. We empirically validate GEORGE on four datasets, and find evidence that it can reduce hidden stratification on real-world machine learning tasks.

## Broader Impact

The potential real-world impact of GEORGE, the approach we present in this work, is that it would allow machine learning practitioners to both *measure* and *mitigate* hidden stratification without requiring any additional prior information. Concretely, this means that users would be able to leverage clusters identified in Step 1 of GEORGE to measure performance gaps between unlabeled subclasses, and that they would subsequently be able to reduce that subclass performance gap via Step 2 of GEORGE. We hope that GEORGE could serve as a drop-in replacement for standard ERM-based techniques in situations where ensuring good performance across many potentially unknown subclasses is important, as it is simple to implement and can be generically applied: all that is required to apply GEORGE to an existing model is (a) clustering within the representation space of a trained model and (b) retraining using a GDRO objective with the cluster assignments used as groups.

As an example of how GEORGE could be important for meaningful practical applications, we consider our results presented on the ISIC dataset in a real-world context. Naively, the overall AUROC on the ISIC dataset obtained using an ERM-trained model is 0.956, which suggests a high-performing model; however, our clustering (Step 1 of GEORGE) reveals that a large fraction of the benign images contain a "spurious" brightly colored patch, which makes them very easy to classify. The model performs substantially worse for cases without such a patch, and worse still on cases for which a clinician would also have required a histopathology examination to make a diagnosis. Thus, if deployed in practice with a target sensitivity value in mind, the appropriate way to set an operating point for this model is in fact *cluster-dependent*; if a single operating point were set using the aggregate ROC curve, the true sensitivity on the histopathology subclass would be substantially lower than intended. This means that even just *measuring* hidden stratification via Step 1 of GEORGE can provide crucial information that would help avoid spurious false negatives at test time—the worst type of error a medical screening application can make.

As shown in the paper, Step 2 of GEORGE can improve performance on underperforming subclasses. Our approach thus provides additional value via a simple retraining procedure that can reduce the amount of hidden stratification exhibited by the model. While our approach will certainly not provide substantial gains in every possible case—for instance, if performance gaps between subclasses are already minimal—we also do not expect it to cause substantial performance degradation. Indeed, even if the clusters returned by GEORGE are random groupings of points that do not align well with the true subclasses, we still expect a model trained to be robust across such groups to perform similarly to a standard ERM model (as in this case the average per-cluster losses are likely to be close to the overall loss on the superclass). This conclusion is empirically supported by the random-GDRO results of Appendix B, which are generally comparable to ERM.

In summary, we hope that GEORGE will have broader impacts by (a) enabling better measurement of hidden stratification via Step 1, even without knowledge of the subclasses, and (b) potentially improving performance on underserved subclasses with Step 2, with only modest additional effort required compared to normal training procedures (i.e., clustering + retraining with GDRO). In addition to medical imaging tasks such as ISIC, subclasses could represent a large number of important categories including race, gender, and others on which one would generally want to ensure good performance on all subclasses, rather than optimizing for average performance while underserving certain categories. If successful, GEORGE can help to detect and mitigate such important performance differences before models are deployed in practice. To support these potential impacts, we have released a complete implementation of our code,[3] with an easily usable PyTorch API. We look forward to engaging with the broader community to improve our work and deploy it on real-world applications.

## Acknowledgments and Disclosure of Funding

We thank Arjun Desai, Pang Wei Koh, Shiori Sagawa, Charles Kuang, Karan Goel, Avner May, Esther Rolf, and Sharon Li for helpful discussions and feedback.

We gratefully acknowledge the support of DARPA under Nos. FA86501827865 (SDH) and FA86501827882 (ASED); NIH under No. U54EB020405 (Mobilize), NSF under Nos. CCF1763315 (Beyond Sparsity), CCF1563078 (Volume to Velocity), and 1937301 (RTML); ONR under No.

N000141712266 (Unifying Weak Supervision); the Moore Foundation, NXP, Xilinx, LETI-CEA, Intel, IBM, Microsoft, NEC, Toshiba, TSMC, ARM, Hitachi, BASF, Accenture, Ericsson, Qualcomm, Analog Devices, the Okawa Foundation, American Family Insurance, Google Cloud, Swiss Re, the HAI-AWS Cloud Credits for Research program, the Schlumberger Innovation Fellowship program, and members of the Stanford DAWN project: Teradata, Facebook, Google, Ant Financial, NEC, VMWare, and Infosys. The U.S. Government is authorized to reproduce and distribute reprints for Governmental purposes notwithstanding any copyright notation thereon. Any opinions, findings, and conclusions or recommendations expressed in this material are those of the authors and do not necessarily reflect the views, policies, or endorsements, either expressed or implied, of DARPA, NIH, ONR, or the U.S. Government.

## Footnotes

[1]In Appendix C, we show that GEORGE also outperforms other subclass-agnostic baselines, such as GDRO trained using the *superclasses* as groups.

[2]We note that if reweighting is not applied to the Waterbirds validation/test sets (see Appendix B.2.2 for explanation), the cluster-robust performance is significantly closer to the true robust performance (within 2 accuracy points, for both ERM and GEORGE); true robust performance for GEORGE also increases to 82.6%, while other methods are relatively unaffected by this reweighting.

[3] https://github.com/HazyResearch/hidden-stratification/

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
