[Supplementary Material]

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

[4]We note that the final step of GEORGE—training a model to minimize the maximum per-cluster risk—can also be done when "soft" (probabilistic) cluster labels are given instead of hard assignments; see Appendix D.5.

[5]https://isic-archive.com/api/v1/

[6]https://github.com/activatedgeek/LeNet-5

[7]We experimented with concatenating the UMAP and loss representations, but found this to reduce performance.

[8]In this plot, validation accuracies were not reweighted as specified in Section B.2; if they are reweighted, the cluster-robust accuracy is not as good an approximation of the true robust accuracy as before, yet is still significantly better than overall accuracy. Nevertheless, having training and validation sets with different distributions is realistic in some situations.

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

# Appendix

## A  Related Work

Our work builds on several active threads in the machine learning literature.

**Hidden Stratification**   Our motivating problem is that of hidden stratification, wherein models trained on superclass labels exhibit highly variable performance on unlabeled subclasses [36]. This behavior has been observed in a variety of studies spanning both traditional computer vision [52, 39, 14, 22] and medical machine learning [36, 17, 19, 11]. Of note is the work of [39], who propose that the existence of "distribution shift" at the subclass level may substantially affect measures of test set performance for image classification models on CIFAR-10. They use a simple mixture model between an "easy" and a "hard" subclass to demonstrate how changes that would not be detectable at the superclass level could affect aggregate performance metrics. [8] extend these ideas by developing notions of visual hardness, and suggest that better loss function design would be useful for improving the performance of machine learning models on harder examples. [37] study how to automatically find large, interpretable underperforming data slices in structured datasets.

Our approach is also inspired by the literature on causality and machine learning, and in particular by the common assumption that the data provided for both training and evaluation are independent and identically distributed (IID) [44]. This is often untrue in real-world settings; in particular, classes in real-world datasets are often composed of multiple subclasses, and the proportions of these subclasses may change between training and evaluation settings—even if the overall class compositions are the same. Many of the guarantees from statistical learning theory break down in the presence of such non-IID data [7], suggesting that models trained using traditional Empirical Risk Minimization (ERM) are likely to be vulnerable to hidden stratification. This motivates the use of the maximum (worst-case) per-subclass risk, rather than the overall average risk, as the objective to be optimized.

**Neural Representation Clustering**   The first stage of the technique we propose for addressing hidden stratification relies heavily on our ability to identify latent subclasses via unsupervised clustering of neural representations learned via ERM. This has been an area of substantial recent activity in machine learning, and has provided several important conclusions upon which we build in our work. The work of [32] and [46], for instance, demonstrate the utility of a simple autoencoded representation for performing unsupervised clustering in the feature space of a trained model. While the purpose of these works is often to show that deep clustering can be competitive with semi-supervised learning techniques, the mechanics of clustering in model feature space explored by these works are important for our present study. Indeed, we directly leverage the conclusion of [32] that Uniform Manifold Approximation and Projection (UMAP) [33] works well as a dimensionality reduction technique for deep clustering in the current study.

Further, the fact that work such as [20] directly uses neural representation clustering to estimate the presence of novel classes in a given dataset provides an empirical basis for our approach, which uses a model trained with ERM to approximately identify unlabeled subclasses within each superclass. Similarly, [25] demonstrate excellent semi-supervised image classification performance by maximizing mutual information between the class assignments of each pair of images. Their work demonstrates not only the utility of a clustering-style objective in image classification, but also suggests that overclustering – using more clusters than naturally exist in the data – can be beneficial for clustering deep feature representations in a manner that is helpful for semi-supervised classification.

A related, but different, approach is that of [14], who explicitly attempt to identify subcategories of classes via a graph and SVM-based "subcategory mining" framework in order to improve overall task performance. The subcategory mining algorithm is quite complicated and uses manually extracted features (rather than automatically learned features, e.g., from CNNs); in addition, this work is geared towards improving overall performance, rather than ensuring good performance on *all* subcategories. Nevertheless, it is an important piece of prior literature.

**Distributionally Robust Optimization**   The second stage of our proposed approach depends on our ability to optimize the worst-case classification loss over existing subgroups, otherwise known as the Distributionally Robust Objective (DRO). This formulation draws a clear connection between

our work and the literature on fairness in machine learning [4], which is at least partially concerned with ensuring that trained models do not disadvantage a particular group in practice. While there exist a wide variety of definitions for algorithmic fairness [21, 30, 26], the common idea that models should be optimized such that they respect various notions of fairness is closely related to the DRO formulation.

A multitude of recent studies have explored optimizing the DRO objective in slightly different contexts. [16], for instance, consider the general case of optimizing the worst-case loss over any possible subgroup of the data; while conceptually important, their results do not assume a learned feature representation, and the necessary assumptions are quite restrictive. Perhaps most relevant to the current work is the study of [43], who propose the group DRO algorithm for training classifiers with best worst-case subgroup performance. Crucially, this algorithm demonstrates improved worst-case subclass performance in cases where triplets $(x, y, g)$ are known for every data point, with $x$ is the input data, $y$ is the true label, and $g$ is a true subgroup label. While [43] present preliminary evidence that group DRO can work well in the presence of imperfect $g$, the efficacy of the algorithm in this setting remains functionally unexplored. We leverage the group DRO algorithm as an optimizer for solving the DRO objective with respect to our approximately identified subclasses.

Other relevant techniques include invariant risk minimization, which attempts to train classifiers that are optimal across data drawn from a mixture of distributions (i.e., a non-IID setting) [1]; methods from slice-based learning that learn feature representations optimized for ensuring high performance on specific subsets, or "slices" of the data [10]; mixture-of-experts models, which explicitly handle learning models for multiple different subsets of data [24]; and techniques for building robust classifiers via domain adaptation [50].

While our work is closely related to these directions, a major difference is that we handle the setting where the different subclasses (i.e., groups, environments, slices, etc.) are unidentified.

**Representation Learning with Limited or Noisy Labels**   A final research thread that is closely related to the work presented here focuses on deep representation learning in the absence of ground truth labels. Our methods are similar in spirit to those from weak supervision [34, 38], which focuses on training models using noisy labels that are often provided programmatically. Our work can be seen as analyzing a new form of fine-grained weak supervision for DRO-style objectives, which is drawn from unsupervised clustering of an ERM representation. Another related line of work is representation learning for few-shot learning [48]; however, our work fundamentally differs in the sense that we assume no access to ground truth subclass labels.

Other methods aim to automatically learn classes via an iterative approach. An early work of this type is [6], which uses iterative clustering and ERM training to learn highly effective feature representations for image classification. More recently, [18] used a self-supervised task to learn semantically meaningful features, and then generate labels using an iteratively refining approach. Our work differs from these in that we do assume access to ground truth *superclass* labels—which provide much more information than in the fully-unlabeled setting—and use clustering *within each superclass* to generate approximate labels. In addition, our primary end goal is not accurate identification of the subclasses, but ensuring good worst-case performance among all subclasses.

Finally, other works aim to promote a notion of "diversity" among feature representations by adding different regularizers. In [51], such a regularizer was introduced in the context of latent space models, to better capture infrequently observed patterns and improve model expressiveness for a given size. More recently, [35] introduced a regularizer that aims to promote diversity of the predicted logits. They showed that this method could also lead to estimation of subclasses within a superclass, without requiring subclass labels. However, this work focused on improving *overall* performance, and specifically improvement of knowledge distillation; by contrast, our goal is to improve *robust* performance. Nevertheless, integrating these recent ideas into our work is an interesting avenue for future work, to potentially further improve the feature learning stage.

# B Experimental Details

## B.1 GEORGE Pseudocode

We provide pseudocode for GEORGE in Algorithm 1, to complement the detailed description of our methodology in Section 4.[4] Note that our model class $\mathcal{F}$ (as per the notation in Section 2.2) is a class of neural networks, composed of a "featurizer" module $f_\theta$ and a "linear classification head" $L$ that takes the feature representation to a prediction.

---

**Algorithm 1** "GEORGE"

---

**Input:** Data and superclass labels $(x, y) = \{(x_i, y_i)\}_{i=1}^n$; loss function $\ell(x_i, y_i; \theta)$; featurizer class $\mathcal{F}(\theta)$ parameterized by $\theta \in \mathbb{R}^p$, dimensionality reducer $g$, e.g., UMAP (default: identity)
**Optional input:** Pretrained featurizer $f_\theta$

**if** featurizer $f_\theta$ provided **then**
  **pass**
**else**
  # train model [featurizer $f_\theta$ and linear classification head $L$] to minimize empirical risk, and save featurizer
  $$f_\theta, L \leftarrow \underset{\theta' \in \mathbb{R}^p, L'}{\operatorname{argmin}} \left\{ \frac{1}{n} \sum_{i=1}^n \ell(L' \cdot f_{\theta'}(x_i), y_i) \right\}$$
**end if**
# compute feature vectors
$\{v_i\}_{i=1}^n = g(f_\theta(x_i)).$
**for** $b = 1$ **to** $B$ **do**
  # cluster features of each superclass
  $\{\hat{z}_i\} \leftarrow \text{GET\_CLUSTER\_LABELS}(\{v_i : y_i = b\})$
**end for**
# train final model to minimize maximum per-cluster risk
$$f_{\hat{\theta}}, \hat{L} \leftarrow \underset{\theta' \in \mathbb{R}^p, L'}{\operatorname{argmin}} \left\{ \max_{c \in \{1, \dots, C\}} \frac{1}{n_c} \sum_{i=1}^n \mathbf{1}(\hat{z}_i = c) \ell(L' \circ f_{\theta'}(x_i), y_i) \right\}$$
return $(f_{\hat{\theta}}, \hat{L})$

---

## B.2 Dataset Details

Below, we describe the datasets used for evaluation in more detail. We provide PyTorch dataloaders to support each one in the attached code.

Each dataset contains labeled subclasses; although the GEORGE procedure does not use the subclass labels at any point, we use them to assess how well GEORGE (a) can estimate the subclass labels and (b) can estimate and improve worst-case subclass performance.

We remark that while we evaluate on binary classification tasks in this work, GEORGE can readily be applied in principle to tasks with any amount of superclasses and subclasses.

### B.2.1 U-MNIST

Undersampled MNIST (U-MNIST) is a binary dataset that divides data from the standard MNIST dataset (which has 60,000 training points) into two superclasses: numbers less than five, and numbers greater than or equal to five. Crucially, the "8" subclass is subsampled at 5% of its usual frequency in MNIST. The rarity of the "8" subclass makes this task much more challenging than the default MNIST task, in terms of robust performance. We use data drawn from the original MNIST validation set as our test set; we create a separate validation set of 12,000 points by sampling from the MNIST training set, and use the remainder for training.

(a) U-MNIST            (b) Waterbirds

(c) ISIC (Histopathology)         (d) CelebA

Figure 5: Performance of ERM, Superclass-DRO (SP-DRO), Random-DRO (R-DRO), GEORGE, and Subclass-DRO (SBC-DRO). We also show GEORGE with BiT embeddings (G-BiT) for CelebA; however, G-BiT performed significantly worse on the other datasets.

On the validation (and test) sets, we do not actually undersample the 8's, as this would leave only 50-60 "8" examples; instead, we downweight these examples when computing validation/test accuracies and losses, to mimic the rarity that would be induced by actually undersampling but still allow for more stable accuracy measurements.

### B.2.2 Waterbirds

The Waterbirds dataset (4,795 training points) used in this work was introduced in [43]. Similar to our approach for U-MNIST, Sagawa et al. [43] create more balanced validation and test sets to allow for stable measurements, and downweight the examples from rare subclasses during evaluation; we follow the same procedure.

### B.2.3 ISIC

The dataset from the International Skin Imaging Collaboration (ISIC) website is, at time of writing, comprised of 23,906 images and their corresponding metadata [12]. We extract the ISIC dataset directly from the site's image archive, which is accessible through a public API.[5] We only use images whose metadata explicitly describe them as "benign" or "malignant." We use these descriptors in order to formulate the problem as a binary classification task that classifies images as either normal or abnormal. Other possible descriptors that exist in the image metadata (which we filter out) include "indeterminate," "indeterminate/benign," "indeterminate/malignant," or no description. We created pre-set training, validation, and test splits from these images by sampling uniformly at random without replacement to assign 80% of examples to the training set, 10% to the validation set, and 10% to the test set.

We derive true subclass information from the image metadata. In particular, we observed that an image belongs in the benign patch subclass if and only if it is an image from the SONIC data repository [45]. As is detailed in the paper, we are retroactively able to identify the histopathology subclass through analysis of the diagnosis confirmation type of each image. Images in the histopathology subclass were explicitly mentioned as such—other possible diagnosis confirmation types include

"single image expert consensus," "serial imaging showing no change," "confocal microscopy with consensus dermoscopy," or no confirmation type.

### B.2.4 CelebA

The CelebA dataset [31] is a standard face classification dataset containing over 200,000 examples ($\approx 163,000$ train) of celebrity faces, each annotated with 40 different attributes. The images contain a wide variety of poses, backgrounds, and other variations. The task is to classify faces as "blond" or "not blond," as in [43]. We use the standard (pre-set) train/validation/test splits for this task.

## B.3 Methods

### B.3.1 Baselines

In addition to ERM, we run two additional baseline methods: superclass-GDRO and random-GDRO. Superclass-GDRO minimizes the maximum loss over each *superclass*, i.e., runs GDRO using the superclasses as groups. Since we assume knowledge of the training superclass labels, this does not require additional information at training time. Random-GDRO runs GDRO using *randomly chosen* groups within each superclass, where the groups are chosen to have the same sizes as the true subclasses. Since we do not assume the subclass sizes are known, this is not a method that would be useful in practice; rather, it helps highlight the difference between running GDRO with labels that do not align well with the true subclasses, and running GDRO with labels that do. (We interchangeably refer to subclass-GDRO, superclass-GDRO, and random-GDRO as subclass-DRO, superclass-DRO and random-DRO, respectively.) Results on each dataset are presented in Figure 5.

### B.3.2 ERM Training Details

The first stage of our procedure is to train a model for each application using ERM. The activations of the resulting model are clustered and used in the second stage of our procedure. Inspired by the results of [43], we explored using either a standard ERM model or an ERM model with high regularization for this stage, selecting between the two based on the quality of the resulting clustering as measured by the Silhouette score (an unsupervised metric). Below, we detail the ERM hyperparameter settings for each dataset.

**U-MNIST**   Our U-MNIST model is a simple 4-layer CNN, based on a publicly available LeNet5 implementation;[6] based on this implementation, we fix the learning rate at 2e-3 and use the Adam optimizer. Each model is trained for 100 epochs. Because the original implementation does not specify a weight decay, we search over weight decay values of $[10^{-3}, 10^{-4}, 10^{-5}]$, and choose the setting with highest average validation accuracy over three trials with different random seeds. Our final parameters are recorded in Table 4. Once the weight decay value is selected, we perform ten separate trials with different random seeds for each method.

**Waterbirds**   Our Waterbirds model uses the `torchvision` implementation of a 50-layer Residual Network (ResNet50), initialized with pretrained weights from ImageNet (as done in [43]). We use hyperparameters reported by [43]: weight decay of 1e-4, learning rate of 1e-3, SGD with momentum 0.9, and 300 epochs. We perform ten separate trials.

**CelebA**   Our CelebA model also uses a `torchvision` pretrained ResNet50, as done in [43]. We use the hyperparameters reported by [43]: weight decay of 1e-4, SGD with momentum 0.9, and 300 epochs. However, we train on 4 GPUs instead of 1. (This change does not substantially affect the results; our ERM and subclass-GDRO results are similar to those reported in [43].) We perform three separate trials (due to the comparatively large size of this dataset and stability of the results).

**ISIC**   Our ISIC model also uses a `torchvision` pretrained ResNet50. Models were trained for 20 epochs using SGD with momentum 0.9 (as done in [41]). Because these hyperparameters were unavailable in the literature for this architecture and task, we grid searched over weight decay values in [0.01, 0.001, 0.0001] and learning rates in [0.0005, 0.001, 0.005, 0.01], selecting the values that maximize the overall AUROC on the validation set, averaged over three trials per hyperparameter

setting. We run five trials with the best hyperparameter setting to obtain the representations used in the clustering step.

Rather than measuring accuracy for ISIC, we use the AUROC (area under the receiver operating characteristic curve), as is standard on this task [41] and other medical imaging tasks [36]. The specific metric of interest is the worst *per-benign-subclass* AUROC for classifying between that subclass and the malignant superclass (e.g., benign no-patch vs. malignant AUROC). Typically, models designed to attain high AUROC are trained by minimizing the empirical risk as usual. For our *robust* models, we instead minimize $\max_{c \in \text{benign}} \left\{ \frac{1}{n_c + n_{\text{malignant}}} \sum_x \mathbf{1}(z_i = c \text{ OR } y_i = \text{malignant}) \ell(x_i, y_i; \theta) \right\}$ - in other words, the maximum over all benign subclasses of the "modified" empirical risk where all other benign subclasses are ignored. We do this because the worst-case loss over any *benign or malignant* subclass is not necessarily a good proxy for the worst-case per-*benign*-subclass AUROC. Due to the dataset imbalance (many fewer malignant than benign images), standard ERM models attain 100% accuracy on the benign superclass and much lower accuracy (and higher loss) on the malignant superclass. By contrast, in practice a classification threshold is typically selected corresponding to a target *sensitivity* value.

### B.3.3 Clustering Details

We apply a consistent clustering procedure to each dataset, which is designed to encourage discovery of clusters of varied sizes, while still being computationally efficient. We emphasize that while the clustering procedure outlined below yields adequate end-to-end results on our datasets, optimizing this part of the GEORGE procedure represents a clear avenue for future work. In particular, we use the Silhouette score as a metric to select between feature representations and number of clusters; while this is a serviceable heuristic, it has several flaws (and in the case of BiT embeddings, misleadingly suggests that they are not a suitable representation due to their low Silhouette score).

1. *Dimensionality Reduction*: As recommended by [32], we use UMAP for dimensionality reduction before clustering; clustering is faster when the data is low-dimensional, and we find that UMAP also typically improves the results. As an alternative to UMAP, we also use the component of the representation that is orthogonal to the decision boundary, which we refer to as the "loss component," as a single-dimensional representation; this can improve clustering on datasets, especially when performance on certain subclasses is particularly poor (as discussed further in Appendix D.3).[7] When the loss component is used to identify clusters, we find that applying higher regularization to the initial ERM model further improves clustering quality, as this regularization "pushes examples further apart" along the loss direction, and adopt this convention in our experiments.

   In each experiment, we select the representation and the number of clusters $k$ based on the parameter setting that achieves the highest average per-cluster Silhouette score. (For all experiments, we set the number of UMAP neighbors to 10 and the minimum distance to 0; further information about these hyperparameters can be found in [33].)

   The fact that simply using the "loss component" can yield reasonable results is arguably surprising, as this essentially amounts to just picking the examples that the original network got wrong (or closer to wrong than others). Nevertheless, especially on tasks with severe data imbalances and "spurious features" (e.g., Waterbirds and CelebA), the rare subclasses do tend to be misclassified at far higher rates, so simply picking the misclassified examples can be a crude but effective heuristic.

2. *Global Clustering*: For each superclass, we search over $k \in 2, \ldots, 10$ to find the clustering that yields the highest average Silhouette score, using the dimensionality reduction procedure identified above. We similarly perform a search over clustering techniques ($k$-means, GMM, etc.), and find that GMM models achieve high average Silhouette scores most often in our applications. Given that GMM clustering also aligns with our theoretical analysis, we use this approach for all datasets. We refer to this global clustering as $f_{C,G}$.

3. *Overclustering*: For each superclass, we take the clustering $f_{C,G}$ achieving the highest average Silhouette score, and then split each cluster $c_i$ into $F$ sub-clusters $c_{i1}, \ldots, c_{iF}$,

where $F$ denotes the "overclustering factor" (fixed to 5 for all experiments). For each sub-cluster $c_{ij}$ whose Silhouette score exceeds the Silhouette score of the corresponding points in the original clustering, and which contains at least $s_{min}$ points (for a small threshold value $s_{min}$), the global clustering $f_{C,G}$ is updated to include $c_{ij}$ as a new cluster (and its points are removed from the base cluster $c_i$). The overclustering factor $F$ was coarsely tuned via visual inspection of clustering outputs (without referencing the true subclass labels); the threshold value $s_{min}$ is used to prevent extremely small clusters, as these can lead to instability when training with GDRO and/or highly variable estimates of validation cluster-robust accuracy. (Note: We do not apply overclustering to 1-dimensional representations, as it tends to create strange within-interval splits.)

### B.3.4 BiT Details

As an alternative to representations from a trained ERM model, we explore the use of BiT embeddings [27], as discussed in Section 6.4. We use the ResNet-50 version of BiT embeddings; specifically, BiT embeddings are the activations of the penultimate layer of a network pretrained on massive quantities of image data (see [27] for more details). The remainder of GEORGE proceeds the same as usual: the embeddings are clustered and then the cluster assignments are used in the GDRO objective.

For BiT, we experimented with both clustering the BiT embeddings directly (under the hypothesis that the BiT embedding space itself is a good representation), and clustering after dimensionality reduction with UMAP. We found that both generally performed similarly, although clustering raw embeddings was slightly better (and much better for U-MNIST); thus, we show results for clustering the raw embeddings. Due to the high dimensionality of these embeddings (2048-d), we use $k$-means clustering when clustering the BiT embeddings, although the rest of our procedure remains the same.

We find that BiT embeddings significantly improve the end-to-end robust performance results on CelebA; however, they perform worse than the standard version of GEORGE on all other datasets, indicating that the task-specific information is important for these other tasks to learn a "good" representation that can be clustered to find superclasses. Indeed, we find that on these other tasks, the BiT clustering is worse than clustering the activations of the ERM model, in terms of precision and recall at identifying poorly-performing subclasses. [For example, when BiT embeddings are used on MNIST, the "8"s are never identified as their own cluster.]

Surprisingly, the clustered BiT embeddings uniformly have a much lower Silhouette score than the clustered ERM embeddings, even for CelebA. Thus, our current unsupervised representation and clustering selection technique would not have identified the BiT embeddings as better for CelebA. Improving the representation and clustering selection metric to do a better job at automatically choosing among different representations is an interesting avenue for future work. We note that if a small validation set with *subclass* labels is available, such a set could be used to select between different clusterings by measuring the cluster alignment with the true subclasses, or measuring which representation eventual leads to the best validation robust accuracy; however, in general we do not assume any prior knowledge about the subclasses in this work.

### B.3.5 GDRO Training Details

In the final step of GEORGE, we train a new model (with the same architecture) using the group DRO approach of [43] with weak subclass labels provided by our cluster assignments, and compare to GDRO models trained using (a) superclass labels only (b) random subclass labels and (c) human-annotated subclass labels. Each result we present is an average over multiple separate trials with different random seeds (10 for U-MNIST, 5 for Waterbirds and ISIC, 3 for CelebA). Below, we describe the hyperparameter search procedure for each such model and dataset.

**U-MNIST**  In the case of U-MNIST, we ran a hyperparameter search over weight decay in [1e-2, 1e-3, 1e-4, 1e-5], and $C$ (the group size adjustment parameter from [43]) in [0, 1, 2].We find performance to be fairly insensitive to the hyperparameters, so choose weight decay of 1e-5 and $C = 0$ for simplicity and consistency with ERM.

**Waterbirds**  For Waterbirds, we use hyperparameters provided by [43], so no additional hyperparameter tuning is required. These hyperparameters are presented in Table 4.

| Dataset | Training Procedure | Epochs | Learning Rate | Batch Size | Weight Decay | Group Adj. Parameter |
|---|---|---|---|---|---|---|
| U-MNIST | ERM | 100 | 2e-3 | 128 | 1e-5 | - |
| U-MNIST | Random-DRO | 100 | 2e-3 | 128 | 1e-5 | 0 |
| U-MNIST | Superclass-DRO | 100 | 2e-3 | 128 | 1e-5 | 0 |
| U-MNIST | GEORGE | 100 | 2e-3 | 128 | 1e-5 | 0 |
| U-MNIST | Subclass-DRO | 100 | 2e-3 | 128 | 1e-5 | 0 |
| Waterbirds | ERM | 300 | 1e-3 | 128 | 1e-4 | - |
| Waterbirds | Random-DRO | 300 | 1e-5 | 128 | 1 | 2 |
| Waterbirds | Superclass-DRO | 300 | 1e-5 | 128 | 1 | 2 |
| Waterbirds | GEORGE | 300 | 1e-5 | 128 | 1 | 2 |
| Waterbirds | Subclass-DRO | 300 | 1e-5 | 128 | 1 | 2 |
| ISIC | ERM | 20 | 1e-3 | 16 | 1e-3 | - |
| ISIC | Random-DRO | 20 | 1e-3 | 16 | 1e-3 | 0 |
| ISIC | Superclass-DRO | 20 | 5e-4 | 16 | 1e-3 | 0 |
| ISIC | GEORGE | 20 | 1e-3 | 16 | 1e-3 | 0 |
| ISIC | Subclass-DRO | 20 | 5e-4 | 16 | 1e-2 | 0 |
| CelebA | ERM | 50 | 1e-4 | 128 | 1e-4 | - |
| CelebA | Random-DRO | 50 | 1e-5 | 128 | 0.1 | 3 |
| CelebA | Superclass-DRO | 50 | 1e-5 | 128 | 0.1 | 3 |
| CelebA | GEORGE | 50 | 1e-5 | 128 | 0.1 | 3 |
| CelebA | Subclass-DRO | 50 | 1e-5 | 128 | 0.1 | 3 |

Table 4: Final hyperparameters used in experiments. (Note that for each dataset, all GEORGE runs use the same hyperparameters regardless of whether they use BiT or ERM embeddings.)

**CelebA**  For CelebA, we again use hyperparameters provided by [43], so no additional hyperparameter tuning is required. These are presented in Table 4.

**ISIC**  Each type of ISIC model is hyperparameter searched over the same space as the original ERM model, in addition to searching over group size adjustment parameter $C$ in [0, 1, 2]. We found performance to be fairly insensitive to both. Hyperparameters with highest validation performance were used in the final runs, and are reported in Table 4.

### B.4  Hyperparameters

In Table 4, we present the selected hyperparameters for the final runs of each dataset and method.

### B.5  Miscellaneous

In all result tables, $\pm$ intervals denote 95% confidence intervals (where half-width is calculated as standard deviation times 1.96 divided by the square root of the number of trials); similarly, in all plots, error bars denote 95% confidence intervals computed the same way.

## C  Additional Experimental Results

In this section, we provide additional ablation experiments.

### C.1  GEORGE results

#### C.1.1  U-MNIST

Dimensionality reduction for this dataset used 2 UMAP components and no loss component, as UMAP achieved higher SIL scores. Our clustering procedure consistently identifies a cluster with a high proportion of the low-frequency "8" subclass. As detailed in the main body, we also often observe a small additional cluster with a high concentration of "7"s written with crosses through the main vertical bar (see Figure 8); performance on this subset is low (below 90%), which explains why cluster-robust performance actually underestimates the true subclass performance on U-MNIST.

Figure 6: Per-trial end-to-end performance on the discovered histopathology subclass of ERM, Superclass-DRO (SPC-DRO), Random-DRO (R-DRO), GEORGE, and Subclass-DRO (SBC-DRO) on the ISIC dataset. We observe that GEORGE fairly consistently outperforms the baselines ERM, SPC-DRO, and R-DRO, and often outperforms SBC-DRO.

### C.1.2 Waterbirds

Dimensionality reduction for this dataset used only 1 component (the loss component); this significantly outperformed UMAP both in terms of SIL score and final robust performance. We observe that while our procedure does not yield clusters with absolutely high frequencies of the minority classes (as shown in Table 2), GEORGE still identifies clusters with high enough precision (i.e., high enough proportions of the poorly-performing subclasses) such that the second stage of GEORGE can substantially improve performance on these subclasses.

### C.1.3 ISIC

Dimensionality reduction for this dataset used 2 UMAP components and no loss component. We observe in Figure 6 that on the three trials (0, 3, and 4) where the clustering step identified a subclass with a concentration of histopathology images (i.e., images requiring histopathology followup) that was greater than 0.75, GEORGE either performs comparably to or outperforms patch/non-patch subclass-DRO.

### C.1.4 CelebA

Dimensionality reduction for CelebA (without BiT) used only 1 component (the loss component). We observe that clustering does not do a good job of identifying the subclasses of either superclass; thus, it is not surprising that the default version GEORGE (i.e., without BiT) performs poorly. In fact, GEORGE performs poorly even compared to the non-ERM baselines. By contrast, GEORGE-BiT does significantly better; the clustering on the (nearly balanced) non-blond superclass attains approximately 95% accuracy at distinguishing between men and women, and the clustering on the blond superclass also significantly improves over the default version of GEORGE.

## C.2 Additional Evaluations and Ablations

### C.2.1 Visualizing Clusters

In Figures 8 and 9, we visualize the representations returned by GEORGE, as well as the clusters it finds and representative examples from each cluster.

### C.2.2 Comparing Cluster-Robust Performance and True Robust Performance

In addition to the results of Table 3 which show that the cluster-robust performance is a good approximation for the true robust performance, we find that the cluster-robust performance typically tracks closely with the true robust performance *throughout training* (with the exception of CelebA without BiT clusters). For example, Figure 7 plots the validation cluster-robust accuracy and validation

Figure 7: Overall accuracy, worst-case cluster accuracy, and worst-case true subclass accuracy on the validation set during a training run of GEORGE on Waterbirds. The worst-case cluster accuracy closely tracks the worst-case true subclass accuracy, whereas the overall accuracy is significantly higher.[8]

Figure 8: True subclasses in the feature space of a trained ERM model. Left panel legend colors points by their true subclass, and displays the validation accuracies of the ERM model on that subclass. Middle panel colors points by the cluster index that GEORGE assigns them. Right panel displays randomly selected examples from each cluster. Datasets: U-MNIST (row 1), Waterbirds (row 2), and ISIC (row 3). Note that for Waterbirds, the vertical axis is the "loss component" and the horizontal axis is the UMAP component.

Figure 9: True subclasses in **BiT** embedding space for CelebA, clusters (middle), and examples from selected clusters (right).

true robust accuracy from a randomly selected training run on Waterbirds. Both metrics are quite close to each other throughout training (while the overall accuracy is significantly higher).

### C.2.3 ERM With Access to True Subclass Labels

In order to further demonstrate the performance gains attributable to GEORGE, we run an additional set of ERM experiments, in which the criterion for selecting the best model checkpoint during training

| Dataset | ERM runtime per epoch | Clustering total runtime | GDRO runtime per epoch |
|---|---|---|---|
| U-MNIST | 3s | 8m | 3s |
| ISIC | 105s | 3m | 105s |
| Waterbirds | 18s | 1m | 18s |
| CelebA | 175s | 46m | 210s |

Table 5: Runtimes for different stages of GEORGE. All runtimes are reported on a machine with 8 CPUs and a single NVIDIA V100 GPU, except for CelebA which was run on a machine with 32 CPUs and four NVIDIA V100 GPUs. Note that the clustering runtime depends on the hyperparameters (e.g., UMAP dimension, $k$-means vs. GMM); we thus report the maximum clustering runtime over all hyperparameters evaluated on the dataset.

is *true* robust accuracy as computed on the validation set (rather than overall accuracy). The purpose of this experiment is to evaluate the relative contributions of (a) the GEORGE learning algorithm and (b) the fact that GEORGE is validating against the cluster-robust accuracy, which is a more accurate approximation of true robust accuracy than is the overall accuracy (which ERM validates against). While we see small performance gains on U-MNIST (0.5% absolute accuracy increase) and Waterbirds (7% absolute accuracy increase) when running ERM and validating against the true robust accuracy, GEORGE validated against *cluster-robust* accuracy still demonstrates stronger performance on the two datasets. suggesting that GEORGE contributes more to model performance than just a better validation metric for early stopping alone. (On ISIC and CelebA, all methods perform quite similarly regardless of validation metric.)

### C.2.4 Runtime

In Table 5, we present runtimes for the standard version of GEORGE broken down by stage.

As the default implementation of GEORGE involves first training an ERM model, clustering its activations, and then training a "robust" model, the total runtime is roughly 2-3x long as that of simply training an ERM model. However, this can be substantially reduced by training the second robust model for fewer epochs. On all datasets, we can recover over 70% of the worst-case performance improvement of GEORGE even when we limit the total runtime to 1.3x that of ERM, simply by training for fewer epochs in the second stage. (On U-MNIST, we also adjusted the LR decay schedule so that decay would occur before the end of the shortened training.) Note that the runtime of typical clustering algorithms scales superlinearly in the number of datapoints; while the clustering runtime is usually less than the training time for the datasets we evaluate on, a remedy for larger datasets could be to only use a random subset of the data for clustering (which typically does not significantly worsen the cluster quality). In addition, we did not attempt to optimize the dimensionality reduction and clustering routines themselves.

### C.2.5 Label Noise

We ran experiments in which a fixed percentage of the data of each subclass was randomly given an incorrect superclass label. With a minor modification (discarding small clusters), GEORGE empirically works well in the presence of label noise when the total number of corrupted labels in each superclass is less than the size of the smallest subclass. Up to this noise threshold, GEORGE attains +3 points robust (i.e., worst-case) accuracy on MNIST and +4 points robust AUROC on ISIC compared to ERM. However, ensuring subgroup-level robustness if there is a larger group of "wrong" examples is difficult because differentiating "real" subclasses from noise becomes challenging. Thus, we do not consider Waterbirds as the smallest subclass (waterbirds on land) is only 1% of the data, and similarly the smallest subclass on CelebA (blond males) is only 3% of the data.

In fact, our clustering approach can even be used to help identify incorrectly labeled training examples. First, if a small "gold" set of correctly labeled examples is available, the clustering found on the training data could be evaluated on this gold set; clusters consisting of mostly incorrectly labeled training examples should have very few members in the gold set. If such a "gold" set is not available, the clusters still allow for much more rapid inspection of the data for incorrect labels, since a few representative examples from each cluster can be inspected instead of a brute-force search through all the training images for incorrectly labeled images. Finally, if one has prior knowledge on the frequency of the rarest subclass in the training data, one can simply discard training examples belonging to poorly-performing clusters smaller than this, treating them as incorrectly labeled.

| Method | Metric | Waterbirds | U-MNIST | CelebA (BiT) |
|--------|--------|-----------|---------|--------------|
| GEORGE | SCAA | 86.7 | 97.9 | 91.4 |
|        | AP | .967 | .9986 | .883 |
| ERM | SCAA | 83.8 | 98.2 | 80.8 |
|     | AP | .984 | .9991 | .912 |

Table 6: Per-subclass averaged accuracy (SCAA) is the mean of the accuracies on each subclass. AP denotes the average precision score (which has a maximum of 1).

### C.2.6 Fixing $k$

If the number of clusters $k$ is held fixed (rather than automatically chosen based on Silhouette score), robust performance tends to initially improve with $k$, before decreasing as large values of $k$ cause fragmented clusters that are less meaningful. For example, robust accuracies on U-MNIST using 2, 5, 10, 25, and 100 clusters per superclass are 95.0%, 96.3%, 95.9%, 94.4%, 90.8% respectively. We also observe similar trends on the other datasets.

### C.2.7 Effect of Model Choice on Subclass Recovery

As suggested in Section 4, choosing an appropriate model class $\mathcal{F}$ for the featurizer $f_\theta$ is important. In particular, $\mathcal{F}$ should ideally contain the inverse of the true generative function $g$, in order to recover the latent features $\vec{V}$ from the data $X$. We demonstrate the importance of model architecture on the ability to separate subclasses in the model feature space by comparing the feature representations of two simple networks on a superclass of the U-MNIST dataset (described in Section 6.1). Fig. 10 shows that the choice of model family can strongly affect the learned feature representation of the initial model and its ability to provide useful information about the subclass. On this dataset, the feature space of a simple fully connected network (Fig. 10a) yields substantially less separation between the known subclasses than does that of a simple convolutional network (Fig. 10b), which displays clusters that clearly correspond to semantically meaningful subclasses.

Figure 10: LeNet5 vs. LeNet 300-100 activations on U-MNIST with true subclass labels (superclass "$< 5$"). A simple convolutional network (LeNet5, right) separates the true subclasses well in feature space, while a network consisting only of fully-connected layers (LeNet 300-100, left) does not.

### C.2.8 Additional Classification Metrics

In Table 6, we compare GEORGE and ERM in terms of per-subclass averaged accuracy (SCAA) and average precision. As expected, GEORGE slightly decreases average precision, as it trades off some average-case performance for better worst-case performance, and GEORGE typically increases per-subclass averaged accuracy (except on U-MNIST, where there is a very slight decrease), due to the fact that it significantly improves performance on poorly-performing subclasses while only slightly decreasing performance on other subclasses.

### C.2.9 Empirical Validation of Lemma 1

Recall that Lemma 1 says that if we know the true data distribution, we can estimate the per-subclass loss $R_c$ by the quantity $\tilde{R}_c$, a reweighted average of the losses in superclass $S(c)$ based on the ratio of the posterior likelihood of a point given subclass $c$ to its posterior likelihood given superclass $S(c)$; Lemma 1 bounds their difference in terms of the number of datapoints $n$. In Figure 11, we empirically validate Lemma 1 on a synthetic mixture-of-Gaussian example (in which the true data distribution is indeed known). We generate data in dimension $d = 3$ with two subclasses each containing six subclasses each, and compute $R_c$ and $\tilde{R}_c$ as per the formulas, using varying numbers of samples $n$ to observe the scaling with $n$. We average results over 20 trials; in each trial, new per-subclass

Figure 11: Comparison of theoretical and simulated convergence of $\tilde{R}_c - R_c$.

distributions are randomly sampled and then new datapoints are sampled. Results are shown in Figure 11. As can be observed from the log-log plot, the slope of the line corresponding to the simulated $|\tilde{R}_c - R_c|$ value is very close to that of the predicted rate; the best fit line has a coefficient of $-0.5065$, corresponding to a $O(n^{-0.5065})$ rate, essentially matching Lemma 1's predicted $O(n^{-0.5})$ rate.

# D   Derivations and Proofs

## D.1   Analysis of Example 3.1.

We restate Example 3.1 below:

**Example 3.1**   The binary attribute vector $\vec{Z}$ has dimension 2, i.e., $\vec{Z} = (Z_1, Z_2)$, while only $Z_2$ determines the superclass label $Y$, i.e., $Y = Z_2$. The latent attribute $Z_1$ induces two subclasses in each superclass, each distributed as a different Gaussian in feature space, with mixture proportions $\alpha$ and $1 - \alpha$ respectively. For linear models with regularized logistic loss, as the proportion $\alpha$ of the rare subclasses goes to 0, the worst-case subclass accuracy of ERM is only $O(\alpha)$, while that of GDRO is $1 - O(\alpha)$.

*Proof.*   Specifically, we consider the following distribution setup: $\vec{Z} \in \{-1, +1\}^2$, with $P(\vec{Z} = (-1, -1)) = P(\vec{Z} = (+1, +1)) = \frac{1-\alpha}{2}$, $P(\vec{Z} = (-1, +1)) = P(\vec{Z} = (+1, -1)) = \alpha/2$, and $P(V_1|Z_1) = \mathcal{N}(4Z_1, \alpha^2)$, $P(V_2|Z_1, Z_2) = \mathcal{N}(Z_1 + 3Z_2, \alpha^2)$, and the label $Y = h(Z_1, Z_2)$ simply equals $Z_2$. We assume the observed data $X = (V_1, V_2)$, i.e., the observed data is the same as the "underlying features" $\vec{V}$.

Thus, the superclass $Y = -1$ is made up of a "big" subclass with distribution $\mathcal{N}((-4, -4), \alpha^2\mathbf{I})$ and relative mixture weight $1 - \alpha$ [corresponding to $\vec{Z} = (-1, -1)$], and a "small" subclass with distribution $\mathcal{N}((+4, -2), \alpha^2\mathbf{I})$ and relative mixture weight $\alpha$ [corresponding to $\vec{Z} = (+1, -1)$], where $\mathbf{I}$ denotes the $2 \times 2$ identity matrix. The superclass $Y = +1$ is made up of a "big" subclass with distribution $\mathcal{N}((+4, +4), \alpha^2\mathbf{I})$ and relative mixture weight $1 - \alpha$ [corresponding to $\vec{Z} = (+1, +1)$], and a "small" subclass with distribution $\mathcal{N}((-4, +2), \alpha^2\mathbf{I})$ and relative mixture weight $\alpha$ [corresponding to $\vec{Z} = (-1, +1)$].

For notational simplicity in the following analysis, we will henceforth rename the label $Y = -1$ as $Y = 0$. The prediction of the logistic regression model on a given sample $(x_1, x_2)$ is $\sigma(w_1 x_1 + w_2 x_2) = \sigma(w^T x)$, where $\sigma(x) := \log(\frac{1}{1+e^{-x}})$ denotes the sigmoid function and $w_1, w_2$ are the weights of the model. The decision boundary is the line $w^T x = 0$; examples with $w^T x < 0$ are classified as $Y = 0$, else they are classified as $Y = 1$. [For simplicity of exposition, we assume there is no bias term, and assume that we regularize the norm of the classifier so that $\|w\|_2 \leq R$ for some constant $R$, as changing the parameter norm does not change the decision boundary. Note that neither assumption is necessary, but they serve to simplify the analysis.]

The logistic loss is the negative log-likelihood, which is $-\sum_i \left(y_i \log(\frac{1}{1+e^{-w^T x}}) + (1-y_i)\log(\frac{1}{1-e^{-w^T x}})\right)$. Note that by symmetry, the loss on the two subclasses with $Z_1 = Z_2$ is the same, as is the loss on the two subclasses with $Z_1 \neq Z_2$. Therefore, we focus on the class $Y = 1$. The expected average loss on the $Y = 1$ superclass is $\mathbb{E}_{x|y=1}[-\log(\frac{1}{1+e^{-w^T x}})] = P(Z_1 = 1|Z_2 = 1) \cdot \mathbb{E}_{x|(z_1,z_2)=(1,1)}[-\log(\frac{1}{1+e^{-w^T x}})] + P(Z_1 = -1|Z_2 = 1) \cdot \mathbb{E}_{x|(z_1,z_2)=(-1,1)}[-\log(\frac{1}{1+e^{-w^T x}})]$.

By 1-Lipschitz continuity of the logistic loss and Jensen's inequality,

$$\left| \mathbb{E}_{x|y=1,z=1}[-\log(\frac{1}{1+e^{-w^T x}})] - \mathbb{E}_{x|y=1,z=1}[-\log(\frac{1}{1+e^{-w^T(4,4)}})] \right| \leq$$
$$\mathbb{E}_{x|y=1,z=1}[|w^T x - w^T(4,4)|] =$$
$$\mathbb{E}_{x|y=1,z=1}[|w_1(x_1 - 4)| + |w_2(x_2 - 4)|] = (|w_1| + |w_2|)\mathbb{E}[|b|],$$

where $b$ is an $N(0, \alpha^2)$ random variable. $\mathbb{E}[|b|] = \alpha\sqrt{2/\pi}$; so, the loss on the $Z_1 = 1$ subclass is bounded in the range $-\log(\frac{1}{1+e^{-w^T(4,4)}}) \pm \alpha\sqrt{2/\pi} \cdot \|w\|_2$.

Similarly, the loss on the $Z_1 = -1$ subclass is bounded in the range $-\log(\frac{1}{1+e^{-w^T(-4,2)}}) \pm \alpha\sqrt{2/\pi} \cdot \|w\|_2$. So, the total loss is bounded in $-(1-\alpha)\log(\frac{1}{1+e^{-w^T(4,4)}}) - \alpha\log(\frac{1}{1+e^{-w^T(-4,2)}}) \pm \alpha\sqrt{2/\pi} \cdot \|w\|_2$. When $\alpha$ is sufficiently small, the first term is $\Theta(1)$, while the latter two are $O(\alpha)$ (under the assumption that $\|w\|_2$ is bounded). For a fixed value of $\|w\|_2$, the first term is minimized when $w/\|w\|_2 = (\frac{1}{\sqrt{2}}, \frac{1}{\sqrt{2}})$, so that $w^T(4,4)$ is as large as possible. A $\Theta(\alpha)$-scale perturbation to the direction $w/\|w\|_2$ results in an increase of $\Theta(\alpha)$ to $-(1-\alpha)\log(\frac{1}{1+e^{-w^T(4,4)}})$. Thus, whenever $\alpha$ is sufficiently small, $w/\|w\|_2$ must be $(\frac{1}{\sqrt{2}}, \frac{1}{\sqrt{2}}) + O(\alpha)$ in order to minimize the loss subject to the $\|w\|_2 \leq R$ constraint. In other words, the regularized ERM solution converges to $(w_1, w_2) = (\frac{1}{\sqrt{2}}, \frac{1}{\sqrt{2}})$ as $\alpha \downarrow 0$.

For the $Z_1 = -1$ subclass, $w^T x$ is a normal random variable with mean $-4w_1 + 2w_2$ and variance $\alpha \|w\|_2^2$. When $\alpha$ is sufficiently small and $w/\|w\|_2 = (\frac{1}{\sqrt{2}}, \frac{1}{\sqrt{2}}) + O(\alpha)$, the quantity $-4w_1 + 2w_2$ is negative with magnitude $O(1)$—and thus, since examples with $w^T x < 0$ are classified as $Y = 0$, this means that for sufficiently small $\alpha$ the fraction of the subclass $Z_1 = -1$ classified correctly as $Y = 1$ is only $O(\alpha)$.

By contrast, the GDRO solution minimizes the maximum per-subclass loss. Since each subclass has the same covariance $\alpha^2 \mathbf{I}$, the GDRO decision boundary is the line that separates the superclass means and has maximum distance to any subclass mean. After normalization to have $\|w\|_2 = 1$, this is the line $(-\frac{1}{\sqrt{5}}, \frac{4}{\sqrt{5}})$; the true solution will be some multiple of this (depending on $\alpha$ and $R$), giving rise to the same boundary. As $\alpha \downarrow 0$, the accuracy of this decision boundary is $1 - O(\alpha)$, since the variance of each subclass is $O(\alpha^2 \mathbf{I})$.

$\square$

## D.2 Proofs from Section 5

### D.2.1 Proof of Lemma 1

**Lemma 1.** *Let $R_c$ be the sample average loss of examples in subclass c. Let $w(x, c) := \frac{\mathcal{P}(x|z=c)}{\mathcal{P}(x|y=S(c))}$. Let $\tilde{R}_c$ be the sample average of $w(x_i, c)\ell(f(x_i), y_i)$ over all examples $x_i$ with* superclass *label $y_i = S(c)$. Then $\tilde{R}_c$ is an unbiased estimate of $R_c$, and their difference converges to $0$ at $O(1/\sqrt{n})$.*

*Proof.* Define $\#_{y=k} := \sum_{i=1}^{n} \mathbf{1}(y_i = k)$ and $\#_{z=c} := \sum_{i=1}^{n} \mathbf{1}(z_i = c)$. Using this notation,

$$R_c = \frac{1}{\#_{z=c}} \sum_{i=1, z_i=c}^{n} \ell(x_i, S(c); \theta), \text{ and } \tilde{R}_c = \frac{1}{\#_{y=S(c)}} \sum_{i=1, y_i=S(c)}^{n} \frac{\mathcal{P}(x_i|z_i = c)}{\mathcal{P}(x_i|y_i = S(c))}\ell(x_i, S(c); \theta).$$

First, observe that $\mathbb{E}[\tilde{R}_c] = \mathbb{E}[R_c]$: the expectation of each term in the summation defining $\mathbb{E}[\tilde{R}_c]$ is $\mathbb{E}_{x\sim[\mathcal{P}|y=S(c)]}\left[\frac{\mathcal{P}(x|z=c)}{\mathcal{P}(x|y=S(c))}\ell(x,S(c);\theta)\right] = \int_{\mathbb{R}^d}\frac{\mathcal{P}(x|z=c)}{\mathcal{P}(x|y=S(c))}\ell(x,S(c);\theta)\mathcal{P}(x|y=S(c))\,dx = \int_{\mathbb{R}^d}\mathcal{P}(x|z=c)\ell(x,S(c);\theta)\,dx = \mathbb{E}_{x\sim[\mathcal{P}(x|z=c)]}[\ell(x,S(c);\theta)] = \mathbb{E}[R_c]$, and so $\mathbb{E}[\tilde{R}_c] = \mathbb{E}[R_c]$.

Now, note that $\frac{\mathcal{P}(x|z=c)}{\mathcal{P}(x|y=S(c))} \geq \mathcal{P}(z=c|y=S(c)) \geq \pi$. Thus, assuming $\ell$ is bounded, each term in the summation defining $\tilde{R}_c$ is bounded, and has mean $\mathbb{E}[R_c]$ as argued above; applying Hoeffding's inequality (to bound the probability that a sum of bounded random variables deviates from its mean by more than a specified amount) yields that $\left|\tilde{R}_c - \mathbb{E}[R_c]\right| \leq O(\frac{1}{\sqrt{n}})$ with high probability. Similarly, applying Hoeffding's inequality to $R_c$ yields that $|R_c - \mathbb{E}[R_c]| \leq O(\frac{1}{\sqrt{n}})$ with high probability. $\square$

### D.2.2 Proof of Theorem 1

We restate Theorem 1 below:

**Theorem 1.** *Let $\hat{R}_{robust} := \max_c \hat{R}_c$. Suppose $\ell, f$ are Lipschitz, $f$ has bounded parameters, and $\mathcal{P}(x|z=c)$ is Gaussian and unique for each subclass $c$. Then, if we estimate $\hat{\mathcal{P}}$ using the algorithm from [3], $\hat{f} := \min_{f\in\mathcal{F}}\hat{R}_{robust}(f)$ satisfies $R_{robust}(\hat{f}) - \min_{f\in\mathcal{F}} R_{robust}(f) \leq \tilde{O}(\sqrt{1/n})$ w.h.p.*

Recall that we define $\hat{R}_c$ to be the same as $\tilde{R}_c$ except with weights $\hat{w}(x,c)$ computed from $\hat{\mathcal{P}}$. More precisely, $\hat{R}_c := \frac{1}{\#_{y=S(c)}}\sum_{i=1,y_i=S(c)}^n \hat{w}(x,c)\ell(x_i,S(c);\theta)$ where $\hat{w}(x,c) := \frac{\hat{\mathcal{P}}(x|z=c)}{\hat{\mathcal{P}}(x|y=S(c))}$. [In Appendix D.5, we provide an algorithm to minimize $\hat{R}_{\text{robust}} = \max_c \hat{R}_c$.]

Our strategy to prove Theorem 1 will be as follows. First, we prove a general statement (that holds regardless of the form of the true data distribution) that relates the total variation (TV) between the true per-subclass distributions and the estimated per-subclass distributions to the difference between $\tilde{R}_c$ and the *perturbed* loss $\hat{R}_c$ (Lemma 2). Next, we bound the total variation between the true and estimated per-subclass distributions in the mixture-of-Gaussians case, using the Gaussian mixture learning algorithm from [3]. Finally, we use standard uniform convergence-type results to yield the final high probability bound on the robust risk of the returned model $\hat{f}$.

#### D.2.2.1 Total variation estimation error to error in loss

**Lemma 2.** *Let $\pi_{\min}$ be the minimum true subclass proportion and $\hat{\pi}_{\min}$ be the minimum estimated subclass proportion. Suppose that we have estimated subclass-conditional distributions $\hat{\mathcal{P}}(x|z)$ such that for each subclass $c$, there exists $c'$ such that $\mathrm{TV}(\mathcal{P}(x|z=c),\hat{\mathcal{P}}(x|z=c')) \leq \epsilon$. If $\ell$ is globally bounded by $M$, then for fixed $f$, $|\hat{R}_{c'} - \tilde{R}_c| \leq \frac{2MC}{\min\{\pi_{\min},\hat{\pi}_{\min}\}}\cdot\epsilon + O(\frac{1}{\sqrt{n}})$ with high probability.*

*Proof.* First, we bound $|\hat{w}(x,c') - w(x,c)| = \left|\frac{\hat{\mathcal{P}}(x|z=c')}{\hat{\mathcal{P}}(x|y=S(c))} - \frac{\mathcal{P}(x|z=c)}{\mathcal{P}(x|y=S(c))}\right| = \frac{1}{\mathcal{P}(x|y=S(c))}\left|\frac{\mathcal{P}(x|y=S(c))}{\hat{\mathcal{P}}(x|y=S(c))}\cdot\hat{\mathcal{P}}(x|z=c') - \mathcal{P}(x|z=c)\right|$. By triangle inequality, this is bounded above by $\frac{1}{\mathcal{P}(x|y=S(c))}\left(\left|\frac{\mathcal{P}(x|y=S(c))}{\hat{\mathcal{P}}(x|y=S(c))}\cdot\hat{\mathcal{P}}(x|z=c') - \hat{\mathcal{P}}(x|z=c')\right| + |\hat{\mathcal{P}}(x|z=c') - \mathcal{P}(x|z=c)|\right)$.

Note that by definition $\hat{\mathcal{P}}(x|y=b) = \sum_{i\in S_b}\hat{\pi}_{b,i}\hat{\mathcal{P}}(x|z=c_i)$, where we define $\hat{\pi}_{b,i} := \hat{\mathcal{P}}(z=c_i|y=b)$ for each $i \in S_b$. In words, $\hat{\pi}_{b,i}$ is the "mixture weight" of subclass $i$ within the population of superclass $b$. We can thus bound $\left|\frac{\mathcal{P}(x|y=S(c))}{\hat{\mathcal{P}}(x|y=S(c))} - 1\right|\hat{\mathcal{P}}(x|z=c')$ by $\frac{1}{\hat{\mathcal{P}}(z=c|y=S(c))}\left|\frac{\mathcal{P}(x|y=S(c))}{\hat{\mathcal{P}}(x|y=S(c))} - 1\right|\hat{\mathcal{P}}(x|y=S(c)) = \frac{1}{\hat{\mathcal{P}}(z=c|y=S(c))}\left|\hat{\mathcal{P}}(x|y=S(c)) - \mathcal{P}(x|y=S(c))\right| \leq \frac{1}{\hat{\pi}_{\min}}\left|\hat{\mathcal{P}}(x|y=S(c)) - \mathcal{P}(x|y=S(c))\right|$.

So, $\mathbb{E}[|\hat{w}(x, c') - w(x, c)|] \leq \frac{1}{\hat{\pi}_{\min}} \mathbb{E}\left[\frac{1}{\mathcal{P}(x|y=S(c))} \cdot (|\hat{\mathcal{P}}(x|y = S(c)) - \mathcal{P}(x|y = S(c))| + |\hat{\mathcal{P}}(x|z = c') - \mathcal{P}(x|z = c)|)\right]$

$= \frac{1}{\hat{\pi}_{\min}} \int_{\mathbb{R}^d} \frac{1}{\mathcal{P}(x|y=S(c))} \cdot (|\hat{\mathcal{P}}(x|y = S(c)) - \mathcal{P}(x|y = S(c))| + |\hat{\mathcal{P}}(x|z = c') - \mathcal{P}(x|z = c)|) \cdot$

$\mathcal{P}(x|y = S(c)) \, dx \leq \frac{C}{\hat{\pi}_{\min}}\epsilon$, since $\int_{\mathbb{R}^d} \left|\hat{\mathcal{P}}(x|z = c) - \mathcal{P}(x|z = c)\right| dx =$

$2TV(\hat{P}(x|z = c), \hat{P}(x|z = c')) \leq 2\epsilon$ by assumption, and $|\hat{\mathcal{P}}(x|y = S(c)) - \mathcal{P}(x|y = S(c))|$ is the weighted sum of the total variations between the distributions of each subclass of $S(c)$, of which there are $\leq C - 1$ [assuming there are at least two superclasses since otherwise the "classification problem" is trivial].

$|\hat{R}_{c'} - R_c| \leq \frac{1}{\#_{y=S(c)}} \sum_{i|y_i = S(c)} |\hat{w}(x_i, c_i') - w(x_i, c_i)|\ell(f(x_i), S(c))$. The expectation of each term in the summation is $\leq \frac{2MC}{\hat{\pi}_{\min}}\epsilon$, since the loss is globally bounded by $M$. Finally, applying Hoeffding's inequality yields that $|\hat{R}_{c'} - \tilde{R}_c| \leq \frac{2MC}{\hat{\pi}_{\min}}\epsilon + O(\frac{1}{\sqrt{n}})$ with high probability.

$\square$

### Total variation in estimated per-subclass distributions: Gaussian case

[3] provides an algorithm for estimating mixtures of Gaussians; they show that $\tilde{O}(kd^2/\epsilon^2)$ samples are sufficient to learn a mixture of $k$ $d$-dimensional Gaussians to within error $\epsilon$ in total variation. Concretely, given $n$ samples from a mixture-of-Gaussian distribution $\mathcal{P}$ and the true number of mixture components $k$, the algorithm in [3] returns a $k$-component mixture-of-Gaussian distribution $\hat{\mathcal{P}}$ such that $TV(\mathcal{P}, \hat{\mathcal{P}}) \leq \tilde{O}(\sqrt{1/n})$. To prove Theorem 1, we use this result and bound the overall total variation error in terms of the maximum per-component total variation error. The proof depends on the key lemma stated below (whose proof appears at the end of this section).

In order to apply Lemma 2, we relate the total variation error $\epsilon$ between the mixtures to the total variation error between the individual mixture components; we show that when $\epsilon$ is small enough, then the total variation error between the mixture components is $O(\epsilon)$ as well. We state this formally in Lemma 3 (proved later in this section).

**Lemma 3.** *Let $\mathcal{P}$ and $\hat{\mathcal{P}}$ be two $k$-component Gaussian mixtures, and suppose the $k$ components of $\mathcal{P}$, denoted by $p_1, \ldots, p_k$, are* distinct *Gaussian distributions and all have nonzero mixture weights. Similarly denote the $k$ components of $\hat{\mathcal{P}}$ by $\hat{p}_1, \ldots, \hat{p}_k$. There exists a constant $c(\mathcal{P})$ depending only on the parameters of $\mathcal{P}$ such that for all sufficiently small $\epsilon > 0$, whenever $TV(\mathcal{P}, \hat{\mathcal{P}}) \leq \epsilon$ it is the case that $\max_{1 \leq i \leq k} \min_{1 \leq j \leq k} TV(p_i, \hat{p}_j) \leq c(\mathcal{P}) \cdot \epsilon$.*

In addition, we use the following standard result from learning theory [29] to relate the minimizer of the estimated robust training loss $\hat{R}_{\text{robust}}$ to the minimizer of the true robust training loss $R_{\text{robust}}$.

**Lemma 4.** *Suppose $g(\cdot, \cdot) \in [-B, B]$. Let $f(\theta) := \mathbb{E}_{(x,y)\sim P}[g(x, y; \theta)]$ and let $\hat{f}(\theta) := \frac{1}{n}\sum_{i=1}^{n} g(x_i, y_i; \theta)$, where $\{(x_i, y_i)\}_{i=1}^{n}$ are IID samples from $P$. Suppose $g(\cdot, \cdot\theta)$ is L-Lipschitz w.r.t. $\theta$, so $f(\theta), \hat{f}(\theta)$ are L-Lipschitz. Then with probability $\geq 1 - O(e^{-p})$, we have*

$$\forall \theta \text{ s.t. } \|\theta\| \leq R, \quad |\hat{f}(\theta) - f(\theta)| \leq O\left(B\sqrt{\frac{p\log(nLR)}{n}}\right) \tag{5}$$

### Theorem 1 Proof

Using the preceding lemmas, we will now prove Theorem 1.

*Proof.* First, we estimate $\hat{\mathcal{P}}$ using the Gaussian mixture learning algorithm from [3]. With high probability, this returns a $k$-component mixture-of-Gaussian distribution $\hat{\mathcal{P}}$ such that $TV(\mathcal{P}, \hat{\mathcal{P}}) \leq \tilde{O}(\sqrt{1/n})$. By Lemma 3, if $n$ is large enough then this means that for each subclass $c$ there exists

a subclass $c'$ such that $\mathrm{TV}(\mathcal{P}(x|z=c),\hat{P}(x|z=c')) \leq \tilde{O}(1/\sqrt{n})$. So, applying Lemma 2, we have that for a fixed function $f$, $|\hat{R}_c - \tilde{R}_c| \leq \tilde{O}(1/\sqrt{n})$, for all subclasses $c$ (WLOG we may reorder the components of $\hat{P}$ so that $c = c'$ for all $c$, for notational convenience). By Lemma 1 and triangle inequality, $|\hat{R}_c - R_c|$ is $\tilde{O}(1/\sqrt{n})$ as well. Thus, for any given $f$, $|\hat{R}_{\mathrm{robust}}(f) - R_{\mathrm{robust}}(f)| = |\max_c \hat{R}_c - \max_c R_c| = \tilde{O}(1/\sqrt{n})$ with high probability.

Let $R^*_{\mathrm{robust}}(f)$ denote the true *population* robust loss. Then Lemma 4 says that $|R_{\mathrm{robust}}(f) - R^*_{\mathrm{robust}}(f)|$ with high probability for all $f$ in the hypothesis class $\mathcal{F}$. Similarly, we can use an analogous uniform convergence result to show that $|\hat{R}_{\mathrm{robust}}(f) - R_{\mathrm{robust}}(f)| \leq \tilde{O}(1/\sqrt{n})$ for all $f$ in the hypothesis class with high probability. Thus, by triangle inequality and union bound, $|\hat{R}_{\mathrm{robust}}(f) - R^*_{\mathrm{robust}}(f)| \leq \tilde{O}(1/\sqrt{n})$ for all $f$ in the hypothesis class with high probability, and in particular this holds for the minimizer $\hat{f}$ of $\hat{R}_{\mathrm{robust}}$.

Thus, under the given assumptions, the excess robust generalization risk (i.e., worst-case subclass generalization risk) of the GEORGE model $\hat{f}$ is $\tilde{O}(1/\sqrt{n})$ [which is near-optimal in terms of sample complexity, since $\Omega(1/\sqrt{n})$ is a generic worst-case lower bound even if the subclass labels are known].

Note that a technical requirement of the above argument is that the samples we use to estimate $\hat{\mathcal{P}}$ should be independent from those we use to compute the robust loss; for this to hold, we may randomly sample half of the examples to learn the distribution $\hat{\mathcal{P}}$ (and its mixture components), and then use the other half to minimize the robust loss. This does not change the asymptotic dependency on the number of samples $n$. [In practice, however, we use all examples in both phases, to get the most out of the data.] $\qquad\square$

**Lemma 3 Proof**

Before we prove Lemma 3, we first provide a simple lemma bounding the total variation distance of two Gaussians in terms of the Euclidean distance between their parameters, directly based on the results from [13].

**Lemma 5.** *Let $p$ be a $d$-dimensional Gaussian with mean $\mu$ and full-rank covariance matrix $\Sigma \in \mathbb{R}^{d \times d}$. Let $p'$ be another Gaussian with mean $\mu'$ and covariance $\Sigma'$. Then there exists a constant $c(\mu, \Sigma)$ [i.e., depending only on the parameters of $p$] such that for all sufficiently small $\epsilon > 0$, whenever $\|\mu - \mu'\|_2 \leq \epsilon$ and $\|\Sigma - \Sigma'\|_F \leq \epsilon$ it is the case that $TV(p, p') \leq c(\mu, \Sigma) \cdot \epsilon$.*

*Proof.* The one-dimensional case is shown in Theorem 1.3 of [13]. The higher-dimensional case follows from Theorems 1.1 and 1.2 of [13]. Note that the constant $c$ does not depend on $\epsilon$, although it may depend on $d$. $\qquad\square$

For convenience, we restate Lemma 3 below.

**Lemma 3.** *Let $\mathcal{P}$ and $\hat{\mathcal{P}}$ be two $k$-component Gaussian mixtures, and suppose the $k$ components of $\mathcal{P}$, denoted by $p_1, \ldots, p_k$, are* distinct *Gaussian distributions and all have nonzero mixture weights. Similarly denote the $k$ components of $\hat{\mathcal{P}}$ by $\hat{p}_1, \ldots, \hat{p}_k$. There exists a constant $c(\mathcal{P})$ depending only on the parameters of $\mathcal{P}$ such that for all sufficiently small $\epsilon > 0$, whenever $TV(\mathcal{P}, \hat{\mathcal{P}}) \leq \epsilon$ it is the case that $\displaystyle\max_{1 \leq i \leq k} \min_{1 \leq j \leq k} TV(p_i, \hat{p}_j) \leq c(\mathcal{P}) \cdot \epsilon$.*

*Proof.* The case $k = 1$ is vacuous (since the "mixture" is simply a single Gaussian); so, suppose $k > 1$. Denote the mixture weights of the true distribution $\mathcal{P}$ as $m_1, \ldots, m_k$, and the mean and covariance parameters of the individual distributions in $\mathcal{P}$ as $\mu_1, \ldots, \mu_k, \Sigma_1, \ldots, \Sigma_k$. In other words, for $x \in \mathbb{R}^d$, $\mathcal{P}(x) = \displaystyle\sum_{i=1}^{k} m_i \mathcal{N}_{\mu_i, \Sigma_i}(x)$, where $m_i \in (0, 1)$ and $\mathcal{N}_{\mu, \Sigma}$ denotes the normal density with mean $\mu$ and covariance $\Sigma$. Similarly, denote the mixture weights of the estimated distribution $\hat{\mathcal{P}}$ by $\hat{m}_1, \ldots, \hat{m}_k$, and the mean and covariance parameters of $\hat{\mathcal{P}}$ by $\hat{\mu}_1, \ldots, \hat{\mu}_k, \hat{\Sigma}_1, \ldots, \hat{\Sigma}_k$. For simplicity assume the covariance matrices $\Sigma_i$ of each component in the true distribution are strictly positive definite (although this is not required).

Define $q(m_1', \ldots, m_k', \mu_1', \ldots, \mu_k', \Sigma_1', \ldots, \Sigma_k') = \int_{\mathbb{R}^d} \left| \sum_{i=1}^{k} m_i \mathcal{N}_{\mu_i, \Sigma_i}(x) - \sum_{i=1}^{k} m_i' \mathcal{N}_{\mu_i', \Sigma_i'}(x) \right| dx.$

The domain of $q$ is constrained to have $m_i' \in [0, 1]$ for all $1 \le i \le k$ and $\sum_{i=1}^{k} m_i' = 1$, as well as to have $\Sigma_i'$ be SPD. By definition, $q(\hat{m}_1, \ldots, \hat{m}_k, \hat{\mu}_1, \ldots, \hat{\mu}_k, \hat{\Sigma}_1, \ldots, \hat{\Sigma}_k)$ is simply twice the total variation between $\mathcal{P}$ and $\hat{\mathcal{P}}$.

Note that, since we assumed the mixture components are unique and $m_i \ne 0$ for all $i$, the only global minima of $q$ [where $q$ evaluates to 0, which means that $\mathcal{P}$ and $\hat{\mathcal{P}}$ are the same distribution] are where $(m_{\pi(i)}', \mu_{\pi(i)}', \Sigma_{\pi(i)}') = (m_i, \mu_i, \Sigma_i)$ for all $1 \le i \le k$, for some permutation $\pi$—in other words, when the two distributions have the exact same mixture components and mixture weights up to permutation. Further, it is not hard to see that the $\epsilon$-sublevel sets of $q$ are contained within compact sets when $\epsilon$ is sufficiently small, and therefore $\lim_{\epsilon \to 0} \{ (m_1', \ldots, \mu_1', \ldots, \Sigma_1', \ldots) : q(m_1', \ldots, \mu_1', \ldots, \Sigma_1', \ldots) \le \epsilon \}$ is exactly the set of global minima of $q$. Finally, note that $q$ is continuous on its domain. Thus, for a fixed distribution $\mathcal{P}$, as $\epsilon \to 0$, the set of points such that $q(m_1', \ldots, \mu_1', \ldots, \Sigma_1', \ldots) \le \epsilon$ is the union of sets of radius $O(\delta(\epsilon))$ around each of the global minima of $q$, where $\delta(\epsilon) \to 0$ as $\epsilon \to 0$. In other words, the set of all Gaussian mixtures $\mathcal{P}'$ with $TV(\mathcal{P}, \mathcal{P}') \le \epsilon$ is the set of all mixtures $\mathcal{P}'$ whose parameters $\{m_i', \mu_i', \Sigma_i'\}$ are $O(\delta(\epsilon))$-close to those of the true distribution $\mathcal{P}$, up to permutation. In particular, if $TV(\mathcal{P}, \mathcal{P}') \le \epsilon$, then for each individual Gaussian component $\mathcal{N}_{\mu_i, \Sigma_i}$ in $\mathcal{P}$, there exists a component $\mathcal{N}_{\mu_j', \Sigma_j'}$ in $\mathcal{P}'$ whose parameters are $O(\delta(\epsilon))$-close to it, i.e., $|m_i - m_j| + \left\| \mu_i - \mu_j' \right\|_2 + \left\| \Sigma_i' - \Sigma_j' \right\|_F \le O(\delta(\epsilon)).$

We now argue that $\lim_{\epsilon \to 0} \frac{\delta(\epsilon)}{\epsilon}$ must be a constant (i.e., $\delta$ is $\Theta(\epsilon)$ as $\epsilon \to 0$) in order for the total variation between the two mixtures to be $\le \epsilon$. We do so by Taylor expanding a set of quantities whose magnitudes lower bound the total variation between $\mathcal{P}$ and $\mathcal{P}'$, and showing that these quantities are locally linear in the parameter differences between $\mathcal{P}'$ and $\mathcal{P}$ when these differences are sufficiently small.

By assumption, $2TV(\mathcal{P}, \mathcal{P}') = q(m_1', \ldots, m_k', \mu_1', \ldots, \mu_k', \Sigma_1', \ldots, \Sigma_k') = \int_{\mathbb{R}^d} \left| \sum_{i=1}^{k} m_i \mathcal{N}_{\mu_i, \sigma_i^2}(x) - \sum_{i=1}^{k} m_i' \mathcal{N}_{\mu_i', \sigma_i^{2'}}(x) \right| dx \le 2\epsilon.$ Notice that

$2TV(\mathcal{P}, \mathcal{P}') = \sup_{S \subseteq \mathfrak{M}^d} \int_S \left| \sum_{i=1}^{k} m_i \mathcal{N}_{\mu_i, \sigma_i^2}(x) - \sum_{i=1}^{k} m_i' \mathcal{N}_{\mu_i', \sigma_i^{2'}}(x) \right| dx \ge$

$\sup_{S \subseteq \mathfrak{M}^d} \left| \int_S \left( \sum_{i=1}^{k} m_i \mathcal{N}_{\mu_i, \sigma_i^2}(x) - \sum_{i=1}^{k} m_i' \mathcal{N}_{\mu_i', \sigma_i^{2'}}(x) \right) dx \right|,$ where $\mathfrak{M}^d$ denotes the collection of all measurable subsets of $\mathbb{R}^d$.

Suppose for now that $d = 1$. Then, $\sup_{S \subseteq \mathfrak{M}} \left| \int_S \left( \sum_{i=1}^{k} m_i \mathcal{N}_{\mu_i, \sigma_i^2}(x) - \sum_{i=1}^{k} m_i' \mathcal{N}_{\mu_i', \sigma_i^{2'}}(x) \right) dx \right| \ge$

$\sup_{c_j \in \mathbb{R}} \left| \int_{-\infty}^{c_j} \left( \sum_{i=1}^{k} m_i \mathcal{N}_{\mu_i, \sigma_i^2}(x) - \sum_{i=1}^{k} m_i' \mathcal{N}_{\mu_i', \sigma_i^{2'}}(x) \right) dx \right| = \sup_{c_j \in \mathbb{R}} |h(\mathcal{P}', c_j)|,$ where we de-

fine $h(\mathcal{P}'; c_j) = h(m_1', \ldots, m_k', \mu_1', \ldots, \mu_k', \sigma_1^{2'}, \ldots, \sigma_k^{2'}; c_j) := \sum_{i=1}^{k} m_i \int_{-\infty}^{c_j} \mathcal{N}_{\mu_i, \sigma_i^2}(x) \, dx -$

$\sum_{i=1}^{k} m_i' \int_{-\infty}^{c_j} \mathcal{N}_{\mu_i', \sigma_i^{2'}}(x) \, dx.$ So, $|h(\mathcal{P}'; c_j)|$ is a lower bound on twice the total variation between $\mathcal{P}$ and $\mathcal{P}'$, for any value of $c_j$. Let $\vec{v} \in \mathbb{R}^{3k}$ denote the vector of parameters $(m_1, \ldots, m_k, \mu_1, \ldots, \mu_k, \sigma_1^2, \ldots, \sigma_k^2)$, and similarly for $\vec{v}'$. For ease of notation we write $h(\vec{v}'; c_j)$ interchangeably with $h(\mathcal{P}'; c_j)$ and $h(m_1', \ldots, m_k', \mu_1', \ldots, \mu_k', \sigma_1^{2'}, \ldots, \sigma_k^{2'}; c_j)$.

Assume $TV(\mathcal{P}, \mathcal{P}') \leq \epsilon$, so as argued before $\|\vec{v}' - \pi(\vec{v})\|_2$ is $O(\delta(\epsilon))$ for some permutation $\pi$ and some function $\delta$ with $\lim_{x \to 0} \delta(x) = 0$. [More precisely, $\lim_{\epsilon \to 0} \max_{\vec{v}':TV(\mathcal{P},\mathcal{P}') \leq \epsilon} \min_{\pi:\pi(k)} \|\vec{v}' - \pi(\vec{v})\|_2 = 0$.]

Without loss of generality, we will henceforth simply write $\vec{v}$ in place of $\pi(\vec{v})$. For notational simplicity, let's write $\delta := \|\vec{v}' - \vec{v}\|_2$. As $h$ is smooth around $\vec{v}$, we can Taylor expand $h$ about the point $\vec{v} = \vec{v}'$ [i.e., $\{m_1' = m_1, ..., \mu_1' = \mu_1, ..., \sigma_1^{2'} = \sigma_1^2, ..., \sigma_k^{2'} = \sigma_k^2\}$] to get

$$h(m_1', \ldots, m_k', \mu_1', \ldots, \mu_k', \sigma_1^{2'}, \ldots, \sigma_k^{2'}; c_j) = h(\vec{v}') =$$
$$h(\vec{v}; c_j) + \nabla h(\vec{v}; c_j)^T (\vec{v}' - \vec{v}) + O(\|\vec{v}' - \vec{v}\|_2^2).$$

Note that $h(\vec{v}; c_j) = 0$ (since $\vec{v}$ are the true parameters).

$\nabla h(\vec{v}; c_j)^T = (\frac{\partial}{\partial m_1'} h(\vec{v}; c_j), \ldots, \frac{\partial}{\partial \mu_1'} h(\vec{v}; c_j), \ldots, \frac{\partial}{\partial \sigma_1'} h(\vec{v}; c_j), \ldots)$.

$\frac{\partial}{\partial m_i'} h(\vec{v}; c_j) = \int_{-\infty}^{c_j} p_{\mu_i, \sigma_i^2}(x) \, dx = \frac{1}{2} \text{erfc}\left(\frac{\mu_i - c_j}{\sqrt{2}\sigma_i}\right)$. [$\sigma_i$ denotes the positive root of $\sigma_i^2$.]

$\frac{\partial}{\partial \mu_i'} h(\vec{v}; c_j) = -\frac{m_i e^{-(c_j - \mu_i)^2/(2\sigma_i^2)}}{\sqrt{2\pi}\sigma_i}$.

$\frac{\partial}{\partial \sigma_i^{2'}} h(\vec{v}; c_j) = \frac{m_i e^{-(c_j - \mu_i)^2/(2\sigma_i^2)}(\mu_i - c_j)}{2\sqrt{2\pi}\sigma_i^3}$.

Define $f_i(x) = \frac{1}{2} \text{erfc}\left(\frac{\mu_i - x}{\sqrt{2}\sigma_i}\right)$ for $1 \leq i \leq k$, $-\frac{m_i e^{-(x-\mu_i)^2/(2\sigma_i^2)}}{\sqrt{2\pi}\sigma_i}$ for $k + 1 \leq i \leq 2k$, and $\frac{m_i e^{-(x-\mu_i)^2/(2\sigma_i^2)}(\mu_i - x)}{2\sqrt{2\pi}\sigma_i^3}$ for $2k + 1 \leq i \leq 3k$. So $\nabla h(\vec{v}; c_j)^T = (f_1(c_j), \ldots, f_{3k}(c_j))^T$.

We have $|\nabla h(\vec{v}; c_j)^T \delta + O(\|\delta\|_2^2)| \leq 2\epsilon$ for all $c_j \in \mathbb{R}$. Now, we claim that it is possible to select $c_1, \ldots, c_{3k}$ such that the $3k \times 3k$ matrix with rows $\nabla h(\vec{v}; c_j)^T$ for $1 \leq j \leq 3k$ is nonsingular.

Proof: Suppose that there is a nonzero vector $\vec{w} \in \mathbb{R}^{3k}$ such that $(f_1(x), \ldots, f_{3k}(x))^T w = 0$ for all $x \in \mathbb{R}$. This means that the function $w_1 f_1(x) + \cdots + w_{3k} f_{3k}(x)$ is identically 0. But this is impossible unless $\vec{w} = 0$, as the $f_i$'s form a linearly independent set of functions (which can easily be seen by looking at their asymptotic behavior). Thus, there is no nonzero vector that is orthogonal to every vector in the set $\bigcup_{x \in \mathbb{R}} \{(f_1(x), \ldots, f_{3k}(x))\}$. In particular, this means that we can find $c_1, \ldots, c_{3k} \in \mathbb{R}$ such that the matrix with rows $(f_1(c_j), \ldots, f_{3k}(c_j))$ is nonsingular (i.e., has linearly independent rows).

Call this matrix $A$. So $\|A\delta + \eta\|_\infty \leq 2\epsilon = 2\epsilon \|\mathbb{1}\|_\infty$ where $\eta \in \mathbb{R}^{3k}$ is such that $\|\eta\|_\infty$ is $O(\|\delta\|_2^2) = O(\|\delta\|_\infty^2)$, and $\mathbb{1}$ denotes the vector of all ones in $\mathbb{R}^{3k}$. So $\|\delta\|_\infty - \|A^{-1}\eta\|_\infty \leq \|\delta + A^{-1}\eta\|_\infty \leq 2\epsilon \|A^{-1}\|_\infty \|\mathbb{1}\|_\infty = 2\epsilon \|A^{-1}\|_\infty$, where $\|A^{-1}\|_\infty$ is the induced $\infty$-norm of $A^{-1}$, and the first inequality follows from the reverse triangle inequality. Note that $\|A^{-1}\eta\|_\infty \leq \|A^{-1}\|_\infty \|\eta\|_\infty \leq O(\|\delta\|_\infty^2)$ since $A^{-1}$ is defined independently of $\delta$.

So, $\|\delta\|_\infty - O(\|\delta\|_\infty^2) \leq 2\epsilon \|A^{-1}\|_\infty$, and thus $\|\delta\|_\infty$ [which is, by definition, the maximum error in any parameter $m_1, \ldots, m_k, \mu_1, \ldots, \mu_k, \sigma_1^2, \ldots, \sigma_k^2$ up to permutation] is $O(\epsilon)$. But then, the total variation between each pair of mixture components $\mathcal{N}_{\mu_i, \sigma_i^2}$ and $\mathcal{N}_{\mu_i', \sigma_i^{2'}}$ is also $O(\epsilon)$, by Lemma 5.

Thus, when $d = 1$, if the total variation between the two Gaussian mixtures $\mathcal{P}$ and $\hat{\mathcal{P}}$ is $O(\epsilon)$, the total variation between each mixture component must also be $O(\epsilon)$ [where the big-O notation suppresses all parameters that depend on the true distribution $\mathcal{P}$], as desired. (Note that total variation is always between 0 and 1.)

Now suppose $d > 1$. Similarly to before, we have $2\epsilon \geq 2TV(\mathcal{P}, \mathcal{P}') \geq$
$$\max_{S \subseteq \mathfrak{M}^d} \left| \int_S \left( \sum_{i=1}^k m_i \mathcal{N}_{\mu_i, \Sigma_i}(x) - \sum_{i=1}^k m_i' \mathcal{N}_{\mu_i', \Sigma_i'}(x) \right) dx \right| \geq$$
$$\max_{\vec{c}_j \in \mathbb{R}^d} \left| \int_{-\infty}^{c_{j1}} \int_{-\infty}^{c_{j2}} \cdots \int_{-\infty}^{c_{jd}} \left( \sum_{i=1}^k m_i \mathcal{N}_{\mu_i, \Sigma_i}(x) - \sum_{i=1}^k m_i' \mathcal{N}_{\mu_i', \Sigma_i'}(x) \right) dx \right| = \max_{\vec{c}_j \in \mathbb{R}^d} |h(\mathcal{P}', \vec{c}_j)|,$$

where we define $h(\mathcal{P}'; \vec{c}_j)$ as $\sum_{i=1}^{k} m_i \int_{-\infty}^{c_{j1}} \int_{-\infty}^{c_{j2}} \cdots \int_{-\infty}^{c_{jd}} \mathcal{N}_{\mu_i, \Sigma_i}(x)\, dx$ $-$

$\sum_{i=1}^{k} m'_i \int_{-\infty}^{c_{j1}} \int_{-\infty}^{c_{j2}} \cdots \int_{-\infty}^{c_{jd}} \mathcal{N}_{\mu'_i, \Sigma'_i}(x)\, dx$. Again, we equivalently denote this by $h(\vec{v}'; c_j)$, where $\vec{v}' \in \mathbb{R}^{k+kd+kd(d+1)/2} := (m'_1, ..., \text{vec}(\mu'_1), ..., \text{vec}(\Sigma'_1), ...)$ denotes the parameters collected into a single vector. As before, we Taylor expand about the point $\vec{v}' = \vec{v}$ to get that $|\nabla h(\vec{v}; \vec{c}_j)^T \delta + O(\|\delta\|_\infty^2)| \le 2\epsilon$, where $\delta := \vec{v}' - \vec{v}$.

Now, we compute the entries of $\nabla h(\vec{v}; \vec{c}_j)$:

$$\frac{\partial}{\partial m'_i} h(\vec{v}; \vec{c}_j) = \int_{-\infty}^{c_{j1}} \int_{-\infty}^{c_{j2}} \cdots \int_{-\infty}^{c_{jd}} \mathcal{N}_{\mu_i, \Sigma_i}(x)\, dx =$$

$$\frac{1}{(2\pi)^{d/2}\sqrt{|\det(\Sigma_i)|}} \int_{-\infty}^{c_{j1}} \int_{-\infty}^{c_{j2}} \cdots \int_{-\infty}^{c_{jd}} e^{-(x-\mu_i)^T \Sigma_i^{-1}(x-\mu_i)/2}\, dx.$$

$$\frac{\partial}{\partial (\mu'_i)_a} h(\vec{v}; \vec{c}_j) = \frac{m_i}{(2\pi)^{d/2}\sqrt{|\det(\Sigma_i)|}} \int_{-\infty}^{c_{j1}} \int_{-\infty}^{c_{j2}} \cdots \int_{-\infty}^{c_{jd}} (\Sigma_i^{-1})_a \cdot (x-\mu_i) e^{-(x-\mu_i)^T \Sigma_i^{-1}(x-\mu_i)/2}\, dx,$$

where $(\mu_i)_a$ denotes the $a^{th}$ entry of the vector $\mu_i$ and $(\Sigma_i^{-1})_a$ is the $a^{th}$ row of the matrix $\Sigma_i^{-1}$.

$$\frac{\partial}{\partial (\Sigma'_i)_{ab}} h(\vec{v}; \vec{c}_j) = -\frac{m_i (\Sigma'_i)_{ab}^{-1}}{2(2\pi)^{d/2}\sqrt{|\det(\Sigma_i)|}} \int_{-\infty}^{c_{j1}} \int_{-\infty}^{c_{j2}} \cdots \int_{-\infty}^{c_{jd}} e^{-(x-\mu_i)^T \Sigma_i^{-1}(x-\mu_i)/2}\, dx +$$

$$\frac{m_i}{(2\pi)^{d/2}\sqrt{|\det(\Sigma_i)|}} \int_{-\infty}^{c_{j1}} \int_{-\infty}^{c_{j2}} \cdots \int_{-\infty}^{c_{jd}} [(\Sigma_i^{-1}(x-\mu_i))(\Sigma_i^{-1}(x-\mu_i))^T]_{ab}\, e^{-(x-\mu_i)^T \Sigma_i^{-1}(x-\mu_i)/2}\, dx,$$

where $M_{ab}$ denotes the $(a,b)$ entry of the matrix $M$ (note that all matrices involved in this expression are symmetric).

Once again, this set of $k + kd + kd(d+1)/2$ partial derivatives, considered as functions of $\vec{c}_j$, comprise a linearly independent set of functions, since the $(\mu_i, \Sigma_i)$ pairs are unique. The remainder of the proof proceeds analogously to the $d = 1$ case. $\qquad\square$

## D.3 Subclass Performance Gaps Enable Distinguishing Between Subclasses

In this section, we give simple intuition for why a performance gap between two subclasses of a superclass implies that it is possible to discriminate between the two subclasses in feature space to a certain extent.

For example, suppose the setting is binary classification, and one of the superclasses has two subclasses with equal proportions in the dataset. Suppose we have access to a model whose training accuracy on one subclass is $x$, while its training accuracy on the other subclass is $y$, where $1 \ge x > y \ge 0$.

Of the correctly classified examples, $\frac{x}{x+y} > \frac{1}{2}$ fraction of them are from the first subclass; similarly, of the incorrectly classified examples, $\frac{1-y}{2-x-y} > \frac{1}{2}$ fraction of them are from the second subclass.

This means that if we just form "proxy subclasses" by splitting the superclass into the correctly classified training examples and incorrectly classified training examples, the resulting groups can in fact be a good approximation of the true subclasses! This is illustrated in Figure 12. For instance, suppose $x = 0.9$ and $y = 0.6$. Then $\frac{x}{x+y} = 0.6$ and $\frac{1-y}{2-x-y} = 0.8$ - so, 60% of the examples in the first group are from subclass 1, and 80% of those in the second group are from subclass 2, which is much better than randomly guessing the true subclasses (in which the concentration of each subclass in each guessed group will approach 50% as $n \to \infty$). In the extreme case, if one subclass has accuracy 1 and the other has accuracy 0, then the superclass decision boundary separates them perfectly (no matter their proportions).

Combined with other information, this helps explain why looking at the way each example is classified (such as the loss of the example or related error metrics) can be helpful to discriminate between the subclasses.

Figure 12: A performance gap between subclasses within the same superclass implies a corresponding degree of separation in feature space. Green and red are true subclasses for the superclass which the model predicts as the gray region; the decision boundary for the superclass classification task also approximately separates the subclasses.

Figure 13: "Inherent hardness": the red and blue superclasses overlap, making it impossible to distinguish between them with perfect overall accuracy. The blue superclass has two subclasses; on the leftmost subclass, the classifier can attain perfect accuracy.

### D.4 Inherent Hardness

We define the "inherent hardness" of a (task, function class) pair as the minimum attainable robust error, i.e.,

$$\operatorname*{argmax}_{f \in \mathcal{F}} \ \min_{c \in \{1,\ldots,C\}} \mathbb{E}_{(x,y)|z=c} \left[ \mathbf{1}(f(x) = y) \right],$$

where the function class is denoted by $\mathcal{F}$. (This can be thought of as the "Bayes robust risk.") We allow the function $f$ to be stochastic: i.e., for a given input $x$, it may output a fixed probability distribution over the possible labels, in which case we define $\mathbf{1}(f(x) = y)$ as the probability assigned by $f$ to the label $y$, given input $x$. By definition, the inherent hardness lower bounds the robust error attained by any classifier in $\mathcal{F}$, regardless of how it is trained or how much data is available. The only way to improve robust performance is therefore to either make the model class $\mathcal{F}$ more expressive (i.e., include more functions in $\mathcal{F}$) or to collect new data such that the covariates $x$ include more information that can be used to distinguish between different classes. (Of course, both of these changes would be expected to improve overall performance as well, if sufficient data is available.) Thus, addressing hidden stratification effects caused by "inherent hardness" is beyond the scope of this work. A simple example of an "inherently hard" task (i.e., a task with nonzero "inherent hardness") is in Figure 13; no classifier can get perfect accuracy on every subclass, because the two superclasses overlap and thus it is impossible to distinguish between them in the region of overlap. Nevertheless, it is possible to attain perfect accuracy on *some* subclasses in this example, meaning that there will still be performance gaps between the subclasses.

### D.5 GDRO with soft group assignments

As shown above in Appendix D.2, we can minimize $\max_{c \in [C]} \mathbb{E}_{x \sim \hat{P}_{S(c)}} [\hat{w}(x,c) \ell(x, S(c); \theta)]$ as a surrogate for $\max_{c \in [C]} \mathbb{E}_{x \sim P_c} [\ell(x, S(c); \theta)]$, where $\hat{\mathcal{P}}_{S(c)}$ is the empirical distribution of training examples of superclass $c$ and $\hat{w}(x,c)$ is shorthand for $\dfrac{\hat{\mathcal{P}}(x|z=c)}{\hat{\mathcal{P}}(x|y=S(c))}$. If we define

the density $A_c(x, z) = \hat{\mathcal{P}}_{S(c)}(x)\mathbf{1}(z = c)$, then $\max_{c \in [C]} \mathbb{E}_{x \sim \hat{\mathcal{P}}_{S(c)}}[\hat{w}(x, c)\ell(x, S(c); \theta)] = \max_{c \in [C]} \mathbb{E}_{(x,z) \sim A_c}[\hat{w}(x, z)\ell(x, S(z); \theta)]$. If we now define $\tilde{\ell}(x, z; \theta) := \hat{w}(x, z)\ell(x, S(z); \theta)$, we see that this falls directly within the group DRO framework of [43]. We thus obtain the following algorithm, which is a minor modification of Algorithm 1 of [43]:

---

**Algorithm 2** Modified Group DRO

---

**Input:** Step sizes $\eta_q, \eta_\theta$; empirical per-superclass distributions $\hat{P}_b$ for each superclass $b \in [B]$
Initialize $\theta^{(0)}$ and $q^{(0)}$
**for** $t = 1, \ldots, T$ **do**
    $c \sim \text{Uniform}(1, \ldots, c)$
    $x \sim \hat{P}_{S(c)}$
    $q' \leftarrow q^{(t-1)}$
    $q'_c \leftarrow q'_c \cdot \exp\left(\eta_q \cdot \hat{w}(x, c) \cdot \ell(x, S(c); \theta^{(t-1)})\right)$
    $q^{(t)} \leftarrow q' / \sum_c q'_c$
    $\theta^{(t)} \leftarrow \theta^{(t-1)} - \eta_\theta \cdot q_c^{(t)} \cdot \hat{w}(x, c) \cdot \nabla_\theta \ell(x, S(c); \theta^{(t-1)})$
**end for**

---

The weights $\hat{w}(x, c)$ correspond to "soft labels" indicating the probability a particular example came from a particular superclass; notice that that $\mathbb{E}_{(x,z) \sim A_c}[\hat{w}(x, z)\ell(x, S(z); \theta)]$ depends on every training example in the *superclass* $S(c)$, so each training example is used in multiple terms in the maximization.

Finally, note that if the assumptions (informally: nonnegativity, convexity, Lipschitz continuity, and boundedness) of Proposition 2 in [43] hold for the modified loss $\tilde{l}(x, z; \theta)$, then the convergence guarantees carry over as well, since Algorithm 2 is a specific instantiation of Algorithm 1 from [43]: the convergence rate of Algorithm 2 is $O(\sqrt{1/T})$, where $T$ is the number of iterations. [Specifically, the average iterate achieves a robust loss that is $O(1/\sqrt{T})$ greater than that of the minimizer of the robust loss.]

In our experiments, we found hard clustering to work better than soft clustering; as it also has the advantage of simplicity, all evaluations were performed with hard clustering.