[Reviews · NeurIPS 2020]

Review 1

Summary and Contributions: This paper addresses the problem of hidden stratification in (image) classification: many classes (e.g. “dog”) contain sub-classes (e.g. “golden retriever”), and a model may perform much better on some subclasses than others. If the subclass labels are known, this can be measured by the “robust performance”, which is the worst-case performance over all subclasses. The authors propose a method (GEORGE) meant to reduce the gap between the overall performance and the robust performance without knowledge of the subclass labels. The basic idea is to first train a classifier using the available labels, then cluster the representations to produce pseudo-labels for subclasses, and finally use these pseudo-labels to train GRDO [42] (a prior method to address stratification in the case where subclass labels are known). The paper also provides a data generating process and uses that framework to prove some asymptotic sample complexity results.

Strengths: This paper addresses an important problem in computer vision that deserves more attention, and the proposed technique is interesting and effective. Based on the experiments in the paper, GEORGE provides large gains in robust performance compared to ERM. More broadly, the experimental evaluation is thorough and multifaceted, often including results from multiple runs to demonstrate robustness.

Weaknesses: The clustering step seems somewhat intricate (see supplementary material), and it is not clear how that procedure came about or how robust the performance of GEORGE is to changes in that procedure. It is not clear how relevant the theoretical contributions in Section 5 are in practice - it would be great to include a frank discussion of that point. If they’re not practically relevant, what is the key obstacle there? If they are practically relevant, is there a way to demonstrate that empirically? For instance, is it possible to empirically demonstrate the convergence behavior predicted by Lemma 1?

Correctness: The proposed approach seems reasonable. The theoretical results are correct to the best of my knowledge, but I did not review the proofs in detail so I can’t comment on line-by-line correctness.

Clarity: The paper is well written and the figures are generally clear and informative.

Relation to Prior Work: The paper is well situated with respect to prior work, particularly given the additional related works section in the supplementary materials.

Reproducibility: Yes

Additional Feedback: It would be interesting to see numbers for other common classification metrics, e.g. average precision or class averaged accuracy (maybe using ground truth subclass labels?). This might paint a more comprehensive picture of what is improving and what is getting worse when we switch from ERM to GEORGE. All of the results are for binary classification tasks and one or two subclasses. Is there a natural way to extend this to multiclass classification and/or to many subclasses? Would GEORGE be expected to perform well as the problem complexity increases in this way? Minor comments: The arrowheads in the graphical model in Figure 2(a) are too small, making it hard to see the direction of the edges. ***Post-Rebuttal Comments*** I think the rebuttal was satisfactory. After reading the rebuttal and the other reviews, I will maintain my previous rating in favor of acceptance.


Review 2

Summary and Contributions: This paper proposes a method to both measure and mitigate hidden stratification (even when subclass labels are unknown). Specifically, the authors firstly observe that unlabeled subclasses are often separable in the feature space. Then, they employ the approximate subclass labels as a form of noisy supervision in a distributionally robust optimization objective. Experiments are conducted on four datasets.

Strengths: + The paper is clearly written and easy to follow. + The proposed method technically sounds and the basic idea is interesting. + The theoretical analysis is good. + Experimental results in both main paper and supplementary materials are good.

Weaknesses: There are several typos through this paper. The authors should carefully proofread their paper.

Correctness: The claims and method are correct. Also, the empirical methodology is correct.

Clarity: This paper is clearly written and easy to follow.

Relation to Prior Work: Yes.

Reproducibility: Yes

Additional Feedback:


Review 3

Summary and Contributions: This paper proposed a method named GEOGRE to measure and mitigate hidden stratification consequences without subclass labels. The main reasons for this phenomenon are inherent hardness and imbalance in subclass sizes. Since a high overall accuracy rate is the only evaluation indicator, the model may consistently prefer superclasses while fail to capture visually meaningful finer-grained variation. Specifically, UMAP dimensionality reduction and Gaussian mixture model clustering are primarily used to produce subclass pseudo labels according to the representation space. The second step is training deep model based on those subclass labels to achieve a better worst-case robust performance. Finally, the paper theoretically analyzed and empirically validated the GEORGE performance on four datasets. Experiments demonstrate the effectiveness of the proposed approach.

Strengths: # Advantages: 1. This paper is well written and attached with sufficient supplementary materials, comprising pseudocode, experiment details, code, specific parameters, theoretical derivations, and proofs as so on. 2. The motivation of using cluster pseudo subclass labels and optimize worst-case performance to measure and mitigate hidden stratification problem is novel. And the method seems to work well. 3. The GEORGE procedure is very applied and could be easily added to the normal used ERM model just by clustering with the pretrained ERM and retraining ERM with those generated subclass labels. 4. Experiments are extensive. Four challenging datasets are used to evaluate the effectiveness of the proposed approach. Multiple ablation studies are provided to give insight into the method.

Weaknesses: # Disadvantages: 1. It’s better to highlight your contributions and provide a whole framework figure to illustrate the process more directly. 2. The quality of Figure 2 needs to be improved for a better aesthetic and expressiveness.

Correctness: Yes

Clarity: Yes

Relation to Prior Work: Yes

Reproducibility: Yes

Additional Feedback:


Review 4

Summary and Contributions: This paper first introduces a generative model to explain how the hidden stratification problem can occur. Then, a framework called GEORGE is introduced to mitigate hidden stratification problem without requiring access to subclass labels. Empirical results on four datasets show that GEORGE outpeform ERM.

Strengths: This paper focus the hidden stratification problem of training deep neural networks with only coarse-grained class labels that results in variable performance across different subclasses. Motivated by the observation of feature representation of deep neural networks often capture information about unlabeled subclasses, this paper proposed GEORGE, a two-step method for mitigating hidden stratification. In the first step, GEORGE estimates subclass labels in feature space via Gaussian mixture model clustering. Then in the second step, the estimated subclass labels are used in a distributional robust optimization objecitve to train a robust classifier. The strengths of this work are: 1.This paper is overall complete. It started by introducing a generative model to study how the hidden stratification problem occur. Then, this paper proposed GEORGE, a two-step framework for mitigating the problem. Theoretical analysis of the proposed method is also provided. Extensive experiments are provided to support the proposed method. 2. In the first step, GEORGE trains a standard ERM model and clusters the subclass features. Empirically, the recovered clusters are ofter aligned closely with the true labels. With such estimated subclass labels, GEORGE is then able to train a robust classifier via a group distributional robust optimiation (GDRO) approach. 3. GEORGE can converge to optimal robust risk with the same sample complexity rate as GDRO when recovering true latent features.

Weaknesses: 1. It is good to see a study of the causes of hidden stratification problem before formally proposed GEORGE. According to the paper, there are two main causes, i.e. inherent hardness and data imbalance. However, it is unclear how the proposed GEORGE address these two causes. Any discussion about this would further improve this paper. 2. The computation cost of GEORGE might be high since it is a two-step framework. Any results to show the computation cost?

Correctness: Yes.

Clarity: Yes.

Relation to Prior Work: Yes.

Reproducibility: Yes

Additional Feedback: After reading the rebuttal and other reviews, and I am satisfied with the rebuttal as the additional computational cost is further reduced by modification on the code. Therefore, I agree to raise my score.

[Author Response · NeurIPS 2020]

We are glad that all reviewers appreciated the soundness of our work, the importance of the hidden stratification (HS)
problem we address, and the extensiveness of our evaluations. We thank the reviewers for their thoughtful questions
and helpful feedback to improve our paper, and we will incorporate the responses below into our revision.

**Computational cost (R4)**: R4 asks about the important issue of computational cost. As mentioned in App. C.2.4,
training with GEORGE takes $\approx$2-3x as long as training with ERM on the same hardware, as GEORGE first trains an
ERM model to obtain a feature representation and then trains a second, robust model. To reduce the cost of GEORGE,
one can train the second robust model for fewer epochs, starting from the ERM model instead of from scratch. We
also modified our code to use GPU acceleration during the clustering procedure. These changes reduce the runtime
of GEORGE to 1.3x that of ERM, while maintaining significant gains in worst-case subclass performance [robust
performance] (Table 1). By contrast, simply training an ERM model for longer does not improve robust performance
(as also observed in [43]); thus, GEORGE allows one to trade off runtime for (often large) gains in robust performance.
With tuning of learning rate schedules and other hyperparameters (HPs), GEORGE's cost could be further reduced.

**Causes of HS (R4)**: R4 asks how GEORGE addresses the different causes of HS. GEORGE primarily focuses on
addressing subclass performance gaps that arise from dataset imbalances (unequal fractions of subclasses in the data).
As discussed in Sec. 3.2 and App. D.4, we define "inherent hardness" as the *minimum possible* worst-case subclass
error that any model can achieve. This may be nonzero due to label noise, insufficient model class expressivity, or
insufficient information in the given features to reliably determine the label; by definition, the only way to address
subclass performance gaps caused by inherent hardness is to choose a richer model class or improve the data quality.

**Origins & robustness of clustering approach (R1)**: The goal of GEORGE's clustering step is to recover clusters that
align with the true subclasses as closely as possible. It should satisfy certain desiderata: 1) auto-select the number of
clusters $k$ (as the subclasses may be unknown); 2) be able to identify clusters of very different sizes (as the subclasses
may have differing frequencies). 1) motivates our procedure of searching over $k$ and selecting the $k$ yielding the best
Silhouette (SIL) score (a metric often used to select $k$ [42]). 2) motivates our "overclustering" procedure described in
App. B.3.3: standard methods (e.g. k-means, EM) often fail to identify small clusters, even if they are well-separated,
but after overclustering we typically *do* find these. (One could instead simply fix $k$ to a large value; in App. C.2.6 we
find that this sometimes works, but has the downside of requiring manual specification, and can spuriously split up
larger clusters.) We apply dimensionality reduction (UMAP) as it often improves clustering quality [34] and supports
useful visualizations. We thank R1 for asking about this, and will more clearly motivate the design of our clustering
procedure in the revision. We hope that building on this method may also be of independent interest. We fixed clustering
HPs (e.g. UMAP HPs, overclustering HPs) to be consistent across tasks; tuning per-task would likely further improve
performance. Our results are fairly insensitive (no significant performance drop) to reasonable variation in these HPs.

**Practical implications of theory (R1)**: R1 asks about the practical relevance of our theory (e.g. Lemma 1). A key
practical takeaway is that if the true data distribution is known, we can estimate the true per-subclass loss $R_c$ by the
quantity $\tilde{R}_c$ defined in Lemma 1 (which is computable without requiring subclass labels); Lemma 1 bounds their
difference. In practice, the data distribution is typically unknown and must itself be estimated; we use Lemma 2 to
deal with this approximation error. Following R1's suggestion, we can empirically validate Lemma 1 for a synthetic
mixture-of-Gaussian example (where the data distribution *is* known). From this distribution, we generate varying
amounts of datapoints $n$ and then compute $\tilde{R}_c$ and $R_c$ for each subclass; fitting the exponent to our averaged results
over several random seeds, we find that $|\tilde{R}_c - R_c|$ converges to 0 at $\approx O(n^{-0.506})$, essentially matching Lemma 1's
predicted $O(n^{-0.5})$ rate. We will detail this and additional empirical validations of our theory in the revision.

**Additional metrics (R1)**: We include additional metrics in Table 2 as suggested by R1. In addition to improving robust
accuracy, GEORGE improves acc. averaged per-subclass (SCAA); ERM has slightly higher average precision (AP).

| | | Waterbirds | U-MNIST | CelebA (BiT) |
|---|---|---|---|---|
| Original results | Robust acc. | 82.6 | 96.1 | 86.1 |
| | Runtime ratio w.r.t. ERM | 2.0 | 3.2 | 1.5 |
| Shortened results | Robust acc. | 76.4 | 95.7 | 86.1 |
| | Runtime ratio w.r.t. ERM | 1.3 | 1.3 | 1.2 |
| ERM results | Robust acc. | 60.4 | 94.2 | 41.1 |

Table 1: Top: Original GEORGE results. (For CelebA, we use BiT embeddings [28], so no ERM model is trained first.) Middle: Modified GEORGE results (fewer epochs in second stage, faster clustering). Bottom: ERM.

| Method | Metric | Waterbirds | U-MNIST | CelebA (BiT) |
|---|---|---|---|---|
| GEORGE | SCAA | 89.3 | 98.4 | 91.4 |
| | AP | .951 | .9986 | .883 |
| ERM | SCAA | 84.1 | 98.3 | 80.8 |
| | AP | .983 | .9991 | .912 |

Table 2: Additional classification metrics (ISIC omitted for space). GEORGE improves SCAA, while ERM has higher AP (which is unsurprising as it optimizes for average performance).

**Higher-cardinality tasks (R1)**: R1 asks if GEORGE generalizes to settings with >2 superclasses and/or >2 subclasses
per superclass. Our results apply directly to any number of superclasses, and any number of subclasses per superclass.
Indeed, the U-MNIST task we evaluate on has 5 subclasses per superclass. Our current results across four datasets
provide strong empirical evidence to suggest GEORGE is a promising approach to improve robust performance; we
agree that evaluating on even more complex and multiclass datasets is an important area for further work.

**Figures (R3, R1)**: We will include higher-resolution, more readable figures in the revision. We thank R3 for the
suggestion to provide a figure to illustrate our overall framework in order to improve clarity; we will also include this.

**Typos (R2)**: We thank R2 for raising the issue of typos; we have since carefully gone through the paper to fix all typos.

[Meta-Review · NeurIPS 2020]

This paper initially received scores of 7,7,7, and 5, and after the rebuttal R4 revised up from a 5 to a 6. The reviewers all commented in the discussion that they were satisfied with the responses they received in the rebuttal. The problem of discovering hidden unlabelled subclasses in labelled datasets is an interesting one and relevant to a general machine learning audience. The authors are encouraged to use the suggestions in the reviews to improve the clarity of the paper e.g. R1's comment about the clustering step, fix the typos (R2), and polish the figures (R3, R1).